# MicroRNA-34a dependent regulation of AXL controls the activation of dendritic cells in inflammatory arthritis

Mariola Kurowska-Stolarska[1], Stefano Alivernini[2], Emma Garcia Melchor[1], Aziza Elmesmari[1], Barbara Tolusso[2], Clare Tange[1], Luca Petricca[2], Derek S. Gilchrist[1], Gabriele Di Sante[2], Chantal Keijzer[1], Lynn Stewart[1], Clara Di Mario[2], Vicky Morrison[1], James M. Brewer[1], Duncan Porter[1], Simon Milling[1], Ronald D. Baxter[1,3], David McCarey[3], Elisa Gremese[2], Greg Lemke[4], Gianfranco Ferraccioli[2], Charles McSharry[1] & Iain B. McInnes[1]

Current treatments for rheumatoid arthritis (RA) do not reverse underlying aberrant immune function. A genetic predisposition to RA, such as HLA-DR4 positivity, indicates that dendritic cells (DC) are of crucial importance to pathogenesis by activating auto-reactive lymphocytes. Here we show that microRNA-34a provides homoeostatic control of CD1c$^+$ DC activation via regulation of tyrosine kinase receptor AXL, an important inhibitory DC auto-regulator. This pathway is aberrant in CD1c$^+$ DCs from patients with RA, with upregulation of miR-34a and lower levels of AXL compared to DC from healthy donors. Production of pro-inflammatory cytokines is reduced by *ex vivo* gene-silencing of miR-34a. miR-34a-deficient mice are resistant to collagen-induced arthritis and interaction of DCs and T cells from these mice are reduced and do not support the development of Th17 cells *in vivo*. Our findings therefore show that miR-34a is an epigenetic regulator of DC function that may contribute to RA.

[1] Institute of Infection, Immunity and Inflammation, University of Glasgow, 120 University Place, Glasgow G12 8TA, UK. [2] Division of Rheumatology, Fondazione Policlinico Universitario A. Gemelli, Catholic University of the Sacred Heart, Rome 00168, Italy. [3] NHS Greater Glasgow and Clyde, 1055 Great Western Road, Glasgow G12 0XH, UK. [4] Molecular Neurobiology Laboratory, The Salk Institute for Biological Studies, 10010 North Torrey Pines Road, La Jolla, California 92037, USA. Correspondence and requests for materials should be addressed to M.K.S. (email Mariola.Kurowska-Stolarska@glasgow.ac.uk) or to I.B.M. (email: Iain.McInnes@glasgow.ac.uk).

Dendritic cells (DC) are sentinels of the immune system that initiate and direct the immune response towards foreign antigens while inducing tolerance to self-antigens[1]. Numerous studies have shown that altered DC function can promote breach of tolerance to self-antigens and development of autoimmunity and inflammation[2–4].

Rheumatoid arthritis (RA) is the most common autoimmune polyarthritis, with a lifetime risk of 3.6% in women and 1.7% in men[5,6], causing pain, disability, impaired quality of life and reduced life expectancy. Genetic, epigenetic and epidemiological studies suggest that autoimmunity lies at the core of RA pathogenesis. RA is particularly associated with breach of tolerance to post-translationally modified self-proteins, usually via citrullination or carbamylation, often predating clinical disease by years[7–9]. These early events are initiated by DCs that stimulate auto-reactive T and B cells[8,10,11], which thereafter orchestrate progression to chronic joint inflammation, destruction and comorbidity[12,13]. The majority of current RA treatments target inflammation; however, partial or non-response is common and sustained remission occurs in a minority of patients, representing a considerable unmet clinical need. For example, tumour necrosis factor (TNF), interleukin-6 (IL-6) receptor (IL-6R) and JAK inhibitors reduce inflammation, and slow down radiographic progression, but do not generally mediate restoration of immune homeostasis[14,15]. Moreover, current approaches targeting the adaptive immune response via B cell depletion and co-stimulatory molecule blockade do not seem to promote tolerance[16]. Therefore, restoring homoeostatic DC function to achieve tolerance would be a compelling strategy for treatment of RA.

Accordingly, we sought to dissect molecular pathways that drive the activation of RA DCs. DCs can be broadly categorized into plasmacytoid and myeloid (conventional) lineages. Human myeloid DCs comprise CD141$^+$, CD16$^+$SLAN$^+$, CD16$^+$SLAN$^-$ and CD1c$^+$ cells[17,18]. In addition, 'inflammatory DCs' can differentiate from monocytes infiltrating inflammatory tissues and show phenotypic overlap (for example, CD1c expression) and function with conventional CD1c$^{+\,18}$ DCs, although the latter cells differentiate from a distinct precursor[19]. Human conventional CD1c$^+$ DCs and inflammatory DC populations are critical for T helper type 17 (Th17) responses (for example, via IL-23, IL-6, IL-1β) and pro-inflammatory TNF production, particularly upon TLR7/8 stimulation[20–24]. The activation of DCs is controlled by multiple negative-feedback loops[25–27]. TAM receptor family containing tyrosine kinase receptors Tyro3, AXL and MERTK are critical in switching off TLR ligand-induced DC activation[26]. Recently, a constitutive AXL expression that was associated with a lower level of major histocompatibility class (MHC) class II was identified as a feature of a distinct subpopulation of human conventional CD1c$^+$ DCs[28].

CD1c$^+$ DC from RA patients show activated phenotype. RA synovial fluid contains both conventional CD1c$^+$ and inflammatory CD1c$^+$ cells that can prime naive T cells[29–31]. In addition, these cells support synovial inflammation by producing TNF upon stimulation with TLR7/8 ligands[23,32,33], and human transgenic TLR8 mice develop spontaneous arthritis due to constitutive activation of DCs that drive autoimmunity[34].

Environmental factors such as smoking strongly influence risk of developing RA, suggesting that epigenetic mechanisms are involved in RA pathogenesis[35]. MicroRNAs are epigenetic regulators that fine-tune complex molecular pathways by mediating mRNA degradation or translation inhibition based on 6–8 nucleotide (nt) complementarity sequences between microRNA and the 3'-untranslated region (UTR) of target mRNAs[36]. Aberrations in miRNA control mechanisms contribute to the development of autoimmune diseases,

including arthritis[37–39]. Herein we describe a role for miR-34a in driving deregulation of DC function in inflammatory arthritis.

## Results

**miR-34a is upregulated in CD1c$^+$ DCs from RA patients.** To identify molecular pathways that are deregulated in RA myeloid cells at the site of pathology, we previously generated a transcriptomic signature of RA synovial fluid (SF) myeloid cells and found key upregulation of miR-34a compared to matched cells from blood[40]. As miR-34a is enriched in DCs[41,42], we hypothesised that miR-34a has a regulatory role in DC function and thereby in the pathogenesis of RA. MiR-34a (chromosome 1p36) and family-members miR-34b/c (co-transcribed from 11q23) have similar functions but have distinct expression patterns[43]. To investigate the expression of miR-34a and c in RA CD1c$^+$ cells, we purified CD1c$^+$ DCs (sorting strategy in Supplementary Fig.1) from peripheral blood (PB) of healthy donors and from PB and SF from RA patients with established RA ($>2$ years from diagnosis). Blood CD1c$^+$ DCs from RA patients had greater miR-34a expression than healthy donors, and this was further increased in SF DCs (Fig. 1a). Elevated miR-34a expression was also observed in blood DCs from patients with early RA ($<6$ months from diagnosis) (Fig. 1b). MiR-34c was expressed only at low levels and was not significantly differentially expressed across disease or tissue and was not further examined (Fig. 1a).

DCs primarily function within lymph node structures or the tissue. We therefore investigated the expression of miR-34a in CD1c$^+$ cells from synovial tissue of patients with elevated disease activity (mean ± s.d. DAS28: $5.07 \pm 0.68$) and compared it to the cells from their PB and to PB cells from age-matched healthy donors. MiR-34a was strongly upregulated in CD1c$^+$ sorted from RA synovial tissues compared to RA PB and healthy PB cells (Fig. 1c). This analysis of an independent patient cohort confirmed an increase expression of miR-34a in RA PB CD1c$^+$ compared to age-matched healthy donors PB cells, and revealed that the highest fold change in PB and synovial tissue was among patients resistant-to-treatment and positively correlated with disease duration (r = 0.543; $P = 0.016$; Pearson Correlation Coefficient). An evaluation of miR-34a expression in blood and synovial tissue CD1c$^+$ cells from patients with comparator inflammatory joint disease, psoriatic arthritis (PsA) showed no significant changes in the blood but an increase in expression in synovial tissue DCs although to a lesser degree than in RA ($P = 0.02$; Kruskal–Wallis with Dunn's multiple comparison test; Supplementary Fig. 2). Demographic and clinical information for both sets of RA patients; and PsA patients is presented in Supplementary Tables 1 and 2. Collectively, these data suggest a potential role for miR-34a in the regulation of DCs function in RA.

We next investigated the regulation of miR-34a expression in CD1c$^+$ DCs. MiR-34a was highest in CD1c$^+$ cells from SF and synovial tissues, which contains both conventional CD1c$^+$ and monocyte-derived CD1c$^+$ cells; therefore, we compared the expression of miR-34a in monocytes and monocyte-derived DCs. The copy number of miR-34a in blood monocyte subsets (CD14$^+$CD16$^-$, CD14$^{dim}$CD16$^+$ and CD14$^+$CD16$^+$) from healthy donors was low (Fig. 2a,b), but increased significantly when they were differentiated to inflammatory CD1c$^+$ DCs by granulocyte macrophage colony-stimulating factor (GM-CSF), which was also reflected by a fold-change as compared to monocytes at day 0 (Fig. 2c). The miR-34a levels in these day 7 immature CD1c$^+$ DCs were rapidly downregulated upon maturation with ligands CL097 for intracellular TLR7/8 and lipopolysaccharide (LPS) for surface TLR4 (Fig. 2d). Thus, miR-34a can be induced during GM-CSF-triggered DC

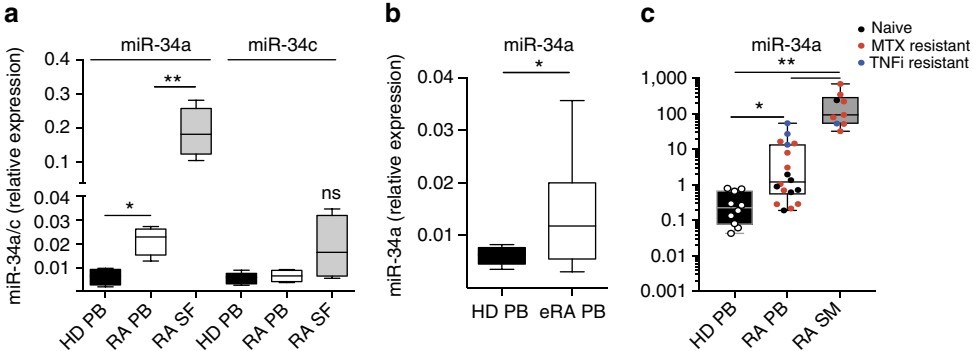

**Figure 1 | MiR-34a expression is upregulated in DCs from patients with RA.** (**a**) CD1c$^+$ DCs were FACS-sorted from PB of healthy donors (HD, $n=6$), and from PB ($n=6$) and SF ($n=6$) of RA patients with established RA of >2 years duration. Compared to HD, miR-34a expression was upregulated in PB CD1c$^+$ in RA patients (*$P<0.05$, Mann–Whitney $U$-test); and further upregulated in matched SF (**$P<0.01$, paired $t$-test). There was no significant difference (ns) in miR-34c expression between these groups. (**b**) miR-34a expression was upregulated in PB CD1c$^+$ cells in patients ($n=14$) with early RA <6 months from diagnosis (eRA), as compared to age-matched HD ($n=6$) (*$P<0.05$, Kolmogorov–Smirnov test). (**c**) MiR-34a expression was upregulated in CD1c$^+$ DCs FACS sorted from dispersed RA synovial membrane (SM, $n=9$) as compared to RA PB ($n=19$) and HD PB ($n=10$) cells (**$P<0.01$ Kruskal–Wallis test and Dunn's multiple comparison test). Data presented as median and inter-quartile range boxplots; and for **c**, each dot represents single sample from naive (naive-to-treatment); MTX resistant (resistant to methotrexate treatment); TNFi resistant (resistant to combined therapy of methotrexate and TNF inhibitors). Demographic and clinical information for samples in **a–b** is presented in Supplementary Table 1 (Cohort 1); and for samples in **c** is presented in Supplementary Table 2 (Cohort 2). ns, nonsignificant.

differentiation and then undergoes dynamic regulation upon DC maturation suggesting a potential role in this process. We next set out to explore the role of miR-34a in DC activation and its impact on arthritis.

**miR-34a$^{-/-}$ mice have reduced antigen-induced arthritis.** We explored a functional role for miR-34a in experimental arthritis using a collagen-induced model in which DCs play a key pathogenic role. Compared to wild-type (WT), miR-34a$^{-/-}$ mice had reduced incidence and severity of arthritis (Fig. 3a,b) accompanied by reduced histopathological articular inflammation and damage (Fig. 3c,d). Serum cytokine profiling revealed selectively reduced IL-17 concentrations (Fig. 3e) suggesting a role for miR-34a in the generation of IL-17-producing, auto-reactive T lymphocytes that drive experimental arthritis[44,45]. There was no difference in concentration of the prototypical Th1 or Th2 cytokines, IFNγ ($P=0.18$; Mann–Whitney $u$-test) and IL-5 ($P=0.08$; Mann–Whitney $u$-test), respectively, nor in the production of collagen-specific antibodies (Supplementary Fig. 3).

To address the specific role of miR-34a in DC-driven antigen-specific T lymphocyte activation, we adoptively transferred naive OVA-specific OT-II T lymphocytes (congenic CD45.1 background) to WT and miR-34a$^{-/-}$ recipients and challenged them with OVA antigen as outlined on Fig. 3f. MiR-34a$^{-/-}$ and WT recipients had similar numbers of total antigen-specific CD4$^+$ and CD4$^+$INFγ$^+$ cells but miR-34a$^{-/-}$ mice showed a selective reduction in antigen-specific Th17 cell numbers (Fig. 3g,h and gating strategy in Supplementary Fig. 4). Collectively, these data suggest that miR-34a expression is important in the development of experimental arthritis, in part via DC-driven expansion of Th17 cells.

**miR-34a regulates DC activation by targeting AXL.** To dissect the specific contribution of miR-34a to DC function in the context of T cell priming and joint inflammation, we evaluated MHC class II, co-stimulatory and inhibitory molecules and a broad range of soluble mediators in miR-34a$^{-/-}$ and control WT bone marrow derived DCs (gating in Supplementary Fig. 5A) stimulated with various TLR ligands. MiR-34a$^{-/-}$ DCs

produced less pro-inflammatory TNF (Fig. 4a), and cytokines implicated in the differentiation (IL-6 and IL-1β) and expansion (p40 subunit of IL-12/IL23 and p19 subunit of IL-23) of Th17 cells (Fig. 4a and Supplementary Fig. 5B). In addition, miR-34a$^{-/-}$ DCs had reduced upregulation of MHC class II surface expression compared to WT DCs upon maturation with TLR ligands. There was no significant difference in the surface expression of co-stimulatory molecules CD86, CD40 and inhibitory PD-L1 molecule (Fig. 4b,c), as well as in expression of candidate adhesion molecules ICAM-1, CD11c, CD11b and CD18a (Supplementary Fig. 6). To test the impact of reduced expression of MHC class II on DC–T-cell interaction, we analysed the contact area between fluorescently labelled OTII T cells (carboxyfluorescein succinimidyl ester (CFSE)-green) and WT or miR-34a$^{-/-}$ DCs (CMTPX-red) in the presence of antigen using a high content image analyser (IN Cell Analyzer 2000) (Fig. 4d). This showed that the increased interaction between DCs and T cells associated with antigen and enhanced with TLR ligand was significantly reduced in miR-34a$^{-/-}$ compared to WT DC co-cultures (Fig. 4d,e).

To extend the studies to the clinical setting, we first over-expressed miR-34a in blood monocyte-derived inflammatory DCs from healthy donors to mimic the constitutively raised miR-34a expression in RA CD1c$^+$ DCs (Fig. 1), and stimulated them with LPS or CL097. MiR-34a over-expression increased DC activation as demonstrated by an increase in their TNF production (Fig. 4f). Consistent with this, inhibition of endogenous miR-34a by gene-silencing during TLR stimulation of monocyte-derived DCs decreased TNF, p40 (IL-12/IL23) and IL-6 production (Fig. 4g). This pro-inflammatory role for miR-34a was confirmed in conventional PB CD1c$^+$ DCs sorted from RA patients in which gene-silencing of miR-34a-inhibited TLR-mediated TNF production (Fig. 4h).

Together these data indicate that miR-34a can regulate DC function, firstly for antigen presentation via modulating class II expression and secondly by promoting pro-inflammatory and Th17 responses. These data were consistent with the reduced development of antigen-specific Th17 response and limited onset and joint pathology of antigen-induced arthritis in miR-34a$^{-/-}$ mice.

To explore mechanisms whereby miR-34a regulated DC activation, we firstly performed *in silico* analysis of predicted

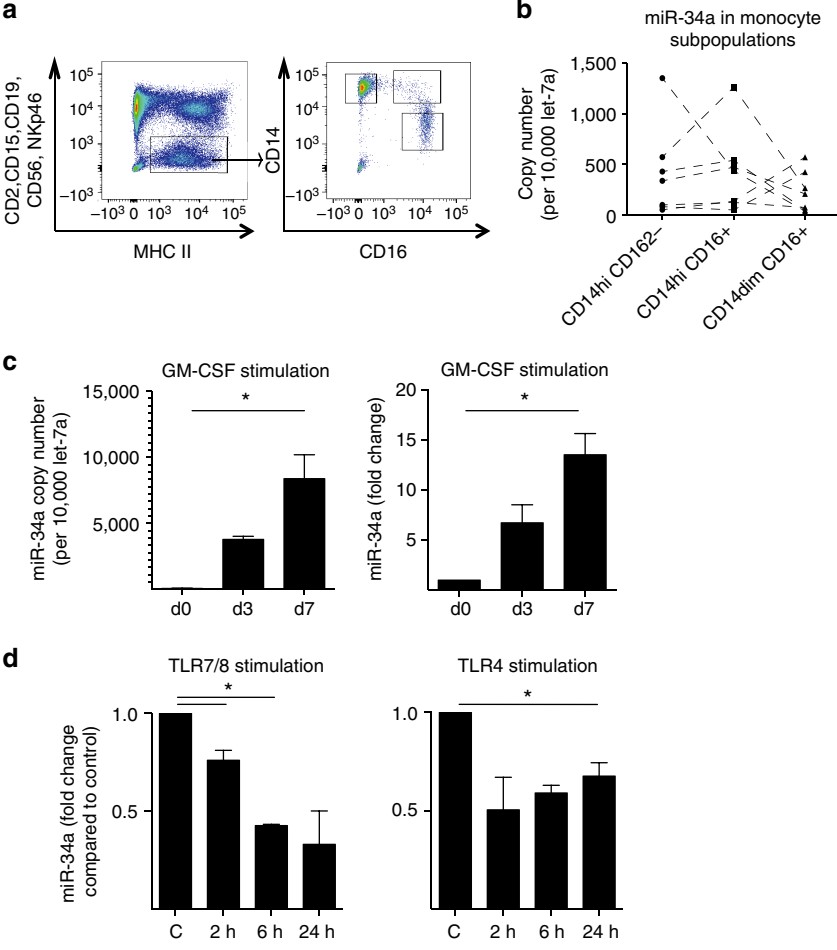

**Figure 2 | MiR-34a expression is regulated by GM-CSF and TLR ligands.** (**a**) A representative FACS-gating strategy for sorting healthy donor human blood monocyte subsets. (**b**) Each monocyte subset expressed low copy numbers of miR-34a relative to let-7a housekeeping control (**c**), which increased following differentiation with GM-CSF (50 ng ml$^{-1}$) for 3 and 7 days. Data in **b** are presented as a dot plot with dotted lines joining the cells from the same donor for clarity. (**d**) Monocyte-derived DCs ($n=3$) were stimulated with TLR agonists CL097 (TLR7/8; 1 μg ml$^{-1}$) or LPS (TLR4; 1 ng ml$^{-1}$) or left un-stimulated (control, C), which decreased miR-34a expression between 2 and 24 h. *$P < 0.05$; one-way ANOVA with Tukey's multiple comparison test. Data are presented as mean ± s.e.m. copy number normalized to let7a or fold-change compared to the respective time point control ($n = 3$-4 biological replicates from three independent experiments).

and validated conserved mouse and human miR-34a mRNA targets (TargetScan, $n = 416$). To map targets to those expressed in cells of myeloid origin we overlapped this with the transcripts present in RA synovial myeloid cells[40]. This strategy identified 43 putative myeloid-cell relevant miR-34a targets (Fig. 5a). This included *AXL*, a tyrosine kinase receptor and member of the TAM receptor family (TYRO3, AXL, MERTK). We chose to investigate AXL because it has been shown to function as a key inhibitory DC auto-regulator by induction of SOCS1/3 and TWIST1 upon binding its ligand Gas6 (refs 26,46–49). We confirmed that Axl-deficient DCs show enhanced activation upon stimulation by TLR ligands manifested by an increased expression of MHC class II and production of IL-6 and TNF compared to WT cells (Supplementary Fig. 7).

We validated whether miR-34a directly targets human *AXL* mRNA using a reporter assay. A luciferase plasmid containing *AXL* 3′-UTR in 'sense' but not in 'anti-sense' orientation showed reduced luciferase activity after transfection with miR-34a mimic (Fig. 5b). Similarly, transfection of human monocyte-derived DCs with miR-34a mimic or inhibitor, repressed or de-repressed *AXL* mRNA expression, respectively (Fig. 5c). To investigate whether the relationship between miR-34a and its target Axl is reciprocal, murine WT bone marrow derived DCs were stimulated with LPS,

and expression levels of both were evaluated at different time points. miR-34a was downregulated by LPS at 24 h, whereas *Axl* mRNA was upregulated but then decreased at later time points, corresponding with the recovery of miR-34a expression (Fig. 5d). Moreover, immature bone marrow derived DCs of miR-34a$^{-/-}$ mice, and human monocyte-derived DCs transfected with miR-34a inhibitor expressed higher levels of AXL protein compared to WT cells, or to cells transfected with control inhibitor, as shown by Western blot (Fig. 5e,f and Supplementary Fig. 8). This was associated with a higher surface expression of Axl in miR-34a$^{-/-}$ DCs in the immature state and during maturation with TLR ligands. Maturation of WT DCs led to an increased surface expression of Axl at 24 h, followed by a decline at 48 h. Expression of Axl in miR-34a$^{-/-}$ DCs was higher than WT cells at all time points (Fig. 5g). This was associated with a decrease in soluble GAS6 levels in miR-34a$^{-/-}$ cultures as compared to controls upon LPS stimulation suggesting that there might be an enhanced binding of GAS6 to increased levels of Axl on miR-34a$^{-/-}$ DCs[50] (Supplementary Fig. 9A). Consistent with these findings, *Axl* and its downstream mediator *Socs-3* (ref. 26) were upregulated while inflammatory mediators *IL-23 (p19)* downregulated in the joint tissue of miR-34a$^{-/-}$ compared to WT mice with experimental arthritis (Fig. 5h,i and

Supplementary Fig. 9B). Together, these data demonstrate that AXL is under the epigenetic control of miR-34a in DCs.

To test whether AXL is responsible for the miR-34a-mediated phenotype of human DC, we transfected human monocyte-derived DCs with miR-34a inhibitor or control inhibitor in the presence of AXL siRNA or control siRNA. Inhibition of miR-34a led to a significant reduction of DC activation, as shown by decreased TNF production, which was restored by downregulation of AXL by

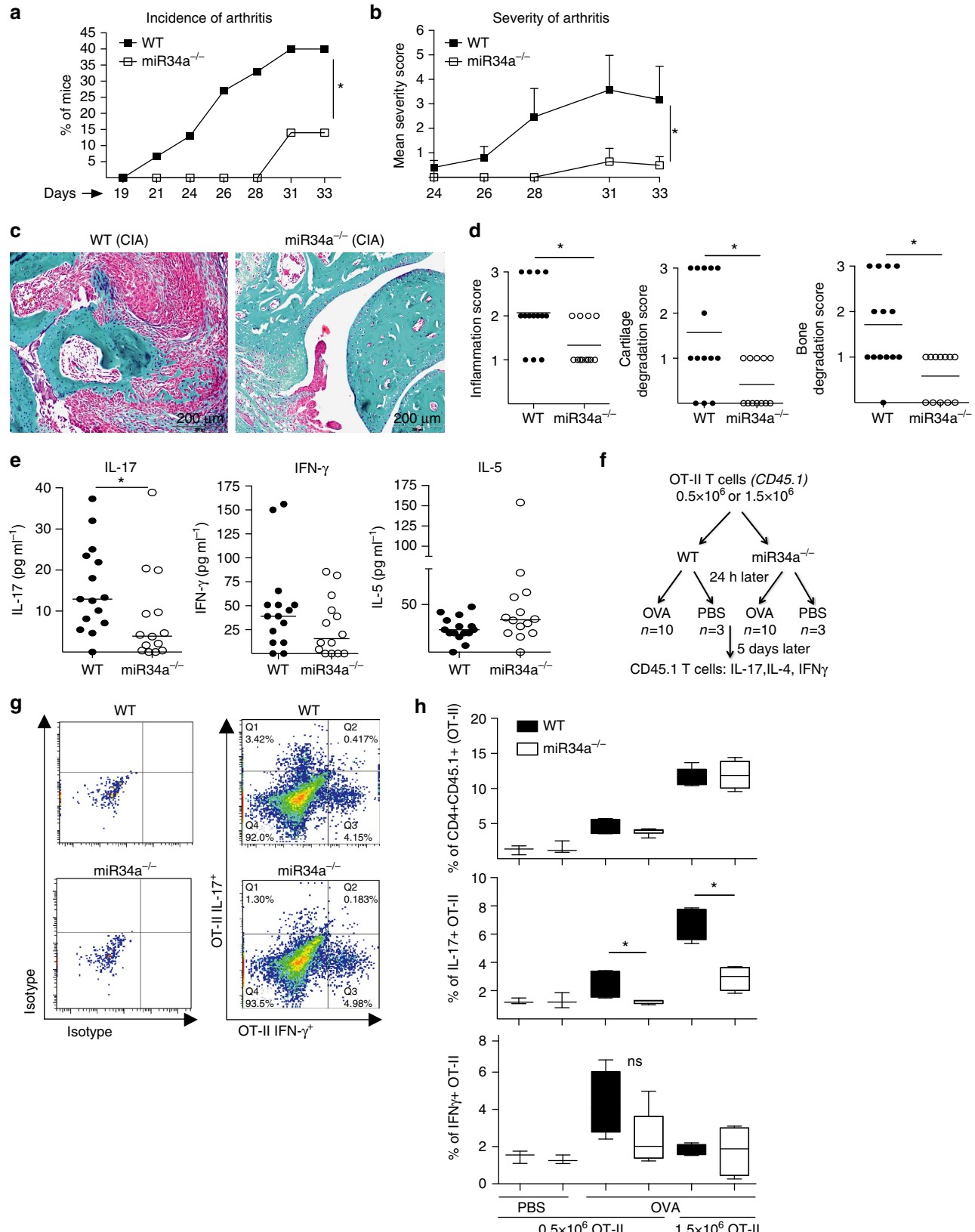

siRNA (Fig. 6a,b). This demonstrates that at least some facets of DC activation regulated by miR-34a are AXL-dependent.

**AXL expression is reduced in RA CD1c$^+$ DCs.** Since CD1c$^+$ DCs of RA patients had high expression levels of miR-34a, we next investigated the mRNA expression of AXL in these CD1c$^+$ cells and found reduced *AXL* expression in two RA cohorts (Fig. 6c,d) that correlated negatively with the expression of miR-34a (r = − 0.463; P = 0.04; Pearson correlation coefficient). The reduced *AXL* expression was associated with an increased activation of RA DCs as shown by an elevated expression level of *IL-6* (r = − 0.560; P = 0.04; Pearson correlation coefficient) (Fig. 6e). This suggests that RA DCs have an epigenetically deregulated homoeostatic mechanism that would normally limit DCs activation preventing inflammation and autoimmunity[26].

We also sought to evaluate other components of the AXL pathway. Serum concentrations of the endogenous ligand of AXL; GAS6, and its decoy receptor soluble AXL (sAXL) which is an inhibitor of the cellular AXL/SOCSs pathway, were measured in RA patient groups: naive-to-treatment (n = 20), resistant-to-treatment (n = 20), in remission (n = 10), and in patients with osteoarthritis (n = 10) and age-matched healthy donors (n = 10). Demographic and clinical information for these cohorts is presented in Supplementary Table 3. We found no difference in serum GAS6 levels in any group of patients compared to healthy donors. The serum concentrations of sAXL in OA were similar to those of healthy donors. In contrast, soluble AXL was upregulated in the naive-to-treatment and resistant-to-treatment RA groups compared to healthy donors (P = 0007 and 0.0042, respectively; Kruskal–Wallis with Dunn's multiple comparison test), with the highest concentration of sAXL detected in the naive-to-treatment group. RA in remission showed a nonsignificant increase in serum concentration of sAXL compared to healthy donors (P = 0.156; Kruskal–Wallis with Dunn's multiple comparison test) (Supplementary Fig. 10). Thus, elevated levels of serum sAXL in RA could be associated with underlying autoimmunity and to a lesser extent inflammation.

In summary, the reduced levels of cell-associated AXL receptor in patients' DCs and higher serum concentrations of GAS6 decoy receptor (sAXL) may prevent engagement of GAS6-AXL inhibitory pathway in RA DCs, particularly in naive-to-treatment early RA.

## Discussion

Current treatments for RA cannot cure damaged joints or re-establish immunological homeostasis[8,15,16]. The genetic loci linked to RA, for example, HLA-DR4 (DRB1*0401) (refs 10,13) suggest that DCs are of crucial importance to pathogenesis by initiating the activation, migration and differentiation of citrullinated protein-specific, auto-reactive Th1, Th17 and follicular helper T cells. These subsequently support the production of auto-antibodies by B cells. Autoimmunity is then followed by localization of the inflammatory response to the joints[8]. The intrinsic biology of DCs encompasses the potential to re-instate immune homeostasis[1]. In support of this concept, a phase I clinical trial demonstrated improvement in DAS28 in RA patients transfused with autologous tolerogenic DCs modified *in vitro* with an NF-kB inhibitor[51], and this was associated with a decrease in antigen-specific effector T cells and serum pro-inflammatory cytokines, and an increase in regulatory T cells. A similar study with intra-articular delivery of autologous tolerogenic DCs showed stabilization of knee symptoms[52]. Thus, a better understanding of the regulatory mechanisms operating in DCs of RA could help development next generation DC-based therapies. We found a negative-feedback control of DC activation, mediated by miR-34a governing immunosuppressive AXL pathway activation, that is aberrant in circulating, SF and synovial tissue CD1c$^+$ DCs of RA patients; and that also contributes to experimental arthritis.

We demonstrated that miR-34a is expressed in conventional human CD1c$^+$ cells, and in monocyte and bone marrow derived inflammatory DCs. Mildner et al.[42] also found its expression in mouse conventional DCs. MiR-34a expression was induced by GM-CSF during differentiation from precursor to inflammatory DCs. Hashimi et al. showed that in these immature human inflammatory DCs, miR-34a supports endocytosis by regulating the JAG1/Wnt1 pathway enhancing antigen uptake[41]. We showed here that during maturation of inflammatory DCs with TLR ligands, that leads to their activation, miR-34a is rapidly downregulated allowing the re-emergence of its epigenetic target AXL which then terminates the activation of DCs. AXL, along with two other members of the TAM receptor family (TYRO3 and MERTK) are key receptors that in response to binding their ligands GAS6 or Protein S produced by innate cells[26] or expressed on the surface of activated T cells (Protein S)[48] terminate both the inflammatory and the adaptive immune response by inducing SOCS1/3 and TWIST1 in order to reinforce homoeostasis. When this is aberrant, for example, in TAM-deficient mice, this leads to spontaneous chronic inflammation and autoimmunity, including production of anti-collagen antibodies and inflammatory histopathology in joints, gut, skin and central nervous system[46–48,53]. The fundamental importance of this regulatory pathway has also been demonstrated in cancer[54] in which reduced expression of miR-34a due to promoter methylation leads to over-expression of AXL thereby promoting an immunosuppressive environment contributing to poor prognosis[55,56]. We found the opposite in RA patients that contributed to RA DC activation thus potentially creating an autoimmune-hypersensitive environment.

In RA patients circulating CD1c$^+$ DCs have constitutively upregulated miR-34a associated with decreased AXL expression. In addition, RA patients have increased serum concentrations of

**Figure 3 | MiR-34a$^{-/-}$ mice are resistant to arthritis.** (a–e) WT (n = 15) and miR-34a$^{-/-}$ (n = 14) mice were sensitized according to the protocol in Methods. Mice were monitored for disease onset and paw swelling from day 10. Mice were killed on day 33 and tissue samples harvested.
(a,b) miR-34a$^{-/-}$ mice had a reduced incidence (a) and severity (b) of arthritis; P < 0.05 two-way ANOVA; severity data are presented as mean ± s.d.
(c,d) miR-34a$^{-/-}$ mice had reduced joint pathology. (c) Representative histology of typical WT and miR-34a$^{-/-}$ joints. Tissue sections (WT, n = 15; miR-34a$^{-/-}$, n = 14) were stained with trichrome (cartilage and bone collagen stained turquoise) and H&E (red; inflammatory synovium).
(d) Quantitative evaluation of synovial inflammation, and cartilage and bone degradation showed reduced scores for miR-34a$^{-/-}$ mice.
(e) MiR-34a$^{-/-}$ CIA mice showed reduced concentrations of IL-17 in the serum compared to WT, but no difference in interferon-gamma (IFNγ) or interleukin (IL)-5. Data are presented as dot-plots with median; (d,e) *P < 0.05, Mann–Whitney u-test. (f) Schematic description of the adoptive transfer of OT-II OVA-specific T cells into WT (n = 13) and miR-34a$^{-/-}$ (n = 13) mice. (g,h) After adoptive transfer of OVA-specific T cells, miR-34a$^{-/-}$ mice showed reduced development of antigen-specific IL-17-producing cells compared to WT recipients. Representative dot-plot representation (g) and quantitative evaluation (h) of total antigen-specific cells and IL-17/IFNγ producing cells in WT and miR-34a$^{-/-}$ recipients are shown. *P < 0.05, Mann–Whitney U-test. Data are presented as median ± IQR obtained in two independent experiments. ns, not significant.

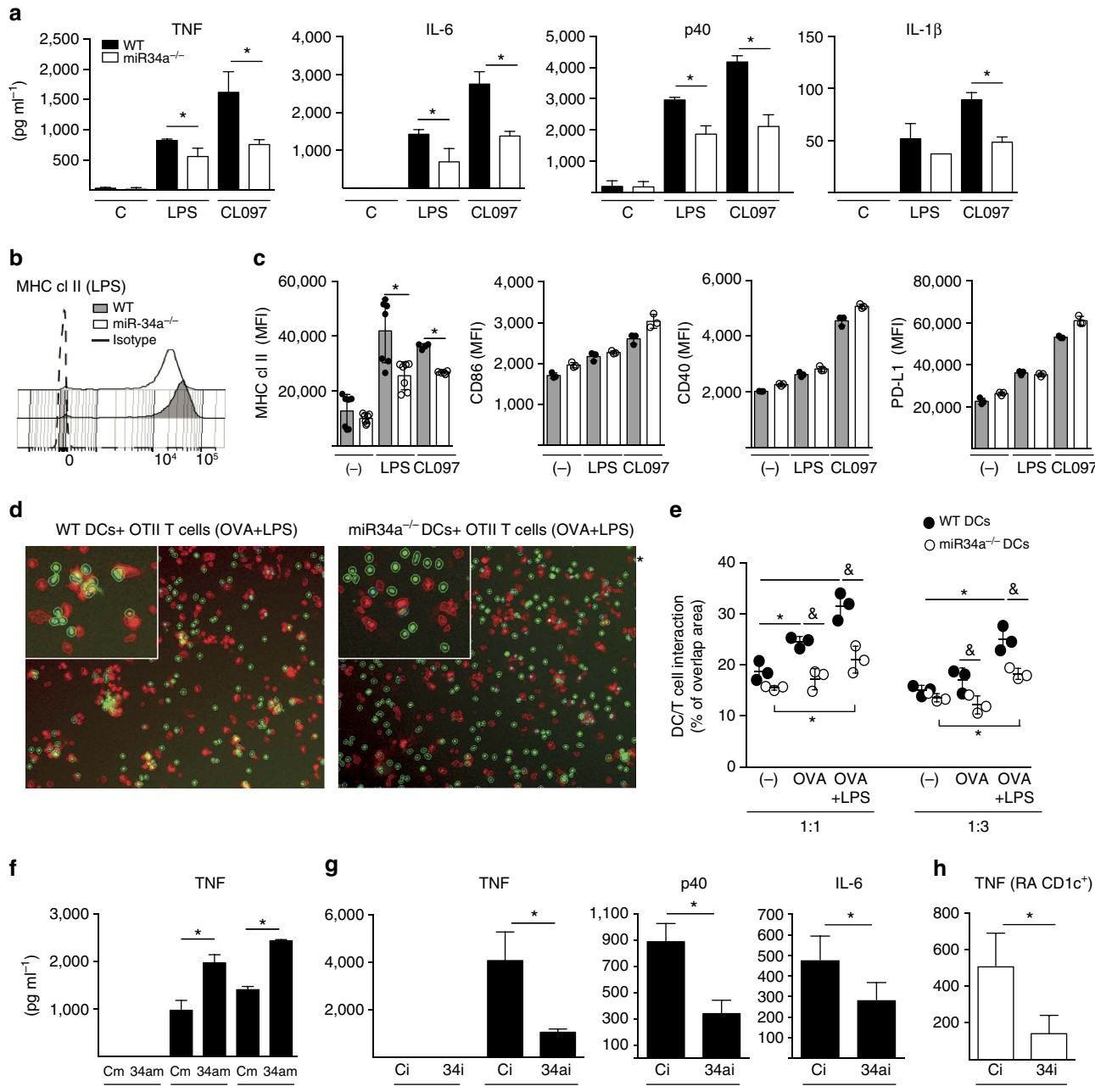

**Figure 4 | MiR-34a drives DC activation.** (**a**) miR-34a$^{-/-}$ DCs show reduced production of pro-inflammatory cytokines. Bone marrow DCs from WT and miR-34a$^{-/-}$ mice ($n = 6$ pooled) were stimulated with LPS or CL097 for 24 h. (**b,c**) miR-34a$^{-/-}$ DCs show reduced expression of MHC class II. WT and miR-34a$^{-/-}$ DCs were stimulated as in **a** for 24 h and the expression of co-stimulatory molecules was evaluated as median fluorescent index (MFI) by flow cytometry. Representative MFI expression of MHC class II (**b**), and quantitative expression of MHC class II, co-stimulatory (CD40 and CD86), and inhibitory (PD-L1) molecules is shown (**c**). *$P < 0.05$ Mann–Whitney $u$-test between genotypes. Data are presented as mean ± s.e.m. of 4–6 replicates from two independent experiments. (**d,e**) miR-34a$^{-/-}$ DCs show a reduced interaction with antigen-specific T cells. CMTPX-labelled WT and miR-34a$^{-/-}$ DCs were co-cultured with CFSE-labelled OT-II CD4$^+$ T cells at 1:1 or 1:3 ratios in the presence of OVA peptide or OVA plus LPS for 24. The contact area was analysed with In Cell Analyzer 2000. Representative pictures of 1:1 cultures (**d**) and quantitative evaluation (**e**) presented as mean ± s.e.m. of % of overlapping areas between DC and T cells compared to the total cell surface area (six technical replicates in total) are shown. *$P < 0.05$ between conditions, $^\&P < 0.05$ between genotypes, two-way ANOVA followed by Tukey's multiple comparison test or unpaired $t$-test. (**f,g**) miR-34a regulates pro-inflammatory cytokine production by human DCs. (**f–h**) Healthy donor monocyte-derived inflammatory DCs ($n = 4$–5) were transfected with miR-34a mimic (34am) or miR-34a inhibitor (34i) or appropriate controls (Cm = mimic or Ci = inhibitor); and (**g**) DCs (conventional CD1c$^+$ DCs; $n = 7$) were sorted from RA patients' blood and transfected with miR-34a inhibitor (34i) or control inhibitor (Ci). The DCs were then stimulated with LPS or CL097 for 24 h. (**f**) Enforced expression of miR-34a increased TNF production; *$P < 0.05$, Kruskal–Wallis followed with Dunn's multiple comparison tests or paired $t$-test. (**g,h**) miR-34a inhibitor reduced the production of cytokines by healthy donor monocyte-derived DCs (**g**) and sorted PB RA CD1c$^+$ DCs (**h**). *$P < 0.05$ paired $t$-test. (**f,g**) Data are presented as mean ± s.e.m of 4–7 biological replicates obtained in separate experiments. ANOVA, analysis of variance.

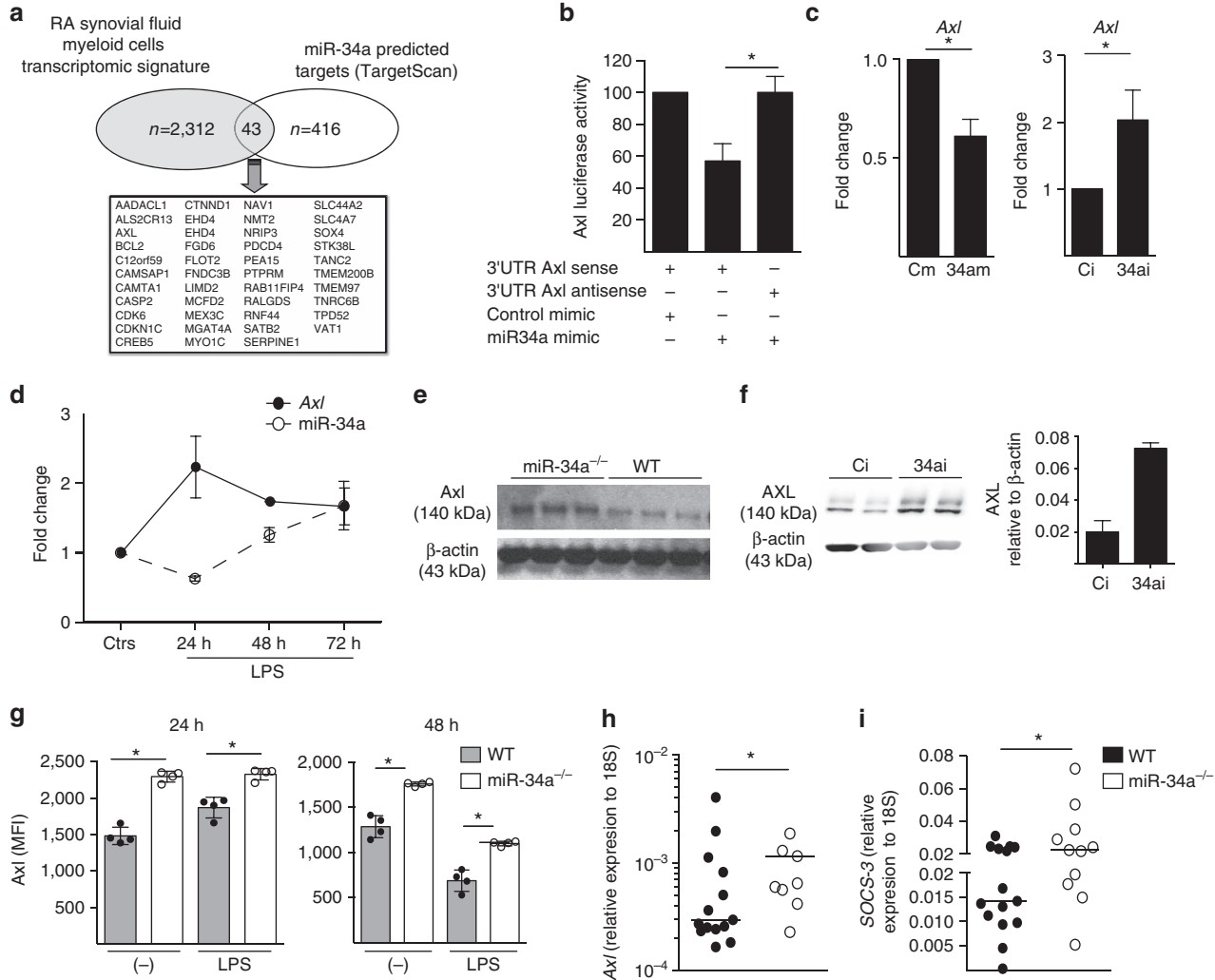

**Figure 5 | MiR-34a controls DC activation by regulating AXL.** (**a**) Epigenetic miR-34a mRNA targets relevant to myeloid cell biology were identified by integrating the conserved miR-34a targets generated by the prediction algorithm TargetScan (Probability of conserved targeting value >0.4) in a Venn diagram along with the mRNA transcriptomic signature of RA SF myeloid cells[68]. (**b,c**) AXL is targeted by miR-34a. (**b**) Luciferase reporter assay for *AXL* shows a reduction in luciferase activity when cells were transfected with miR-34a mimic (details in Methods). Data are presented as mean ± s.e.m of three technical replicates and is representative of two independent experiments. *P<0.05, paired *t*-test between *AXL* sense and anti-sense. (**c**) *AXL* mRNA expression is regulated by miR-34a in monocyte-derived DCs *in vitro*. DCs transfected with either miR-34a mimic (34am) or miR-34a inhibitor (34ai) or appropriate controls (Cm = mimic; Ci = inhibitor), had decreased or increased *AXL* expression, respectively, after 24 h. Data are presented as mean ± s.e.m. of three biological replicates; *P<0.05, Mann–Whitney *U*-test. (**d**) miR-34a downregulation during LPS-induced mouse bone marrow DC maturation is associated with an increase in *Axl* mRNA at 24hr. Data presented as mean ± s.e.m of three technical replicates. (**e**) GM-CSF differentiated mouse miR-34a[−/−] DCs contain higher concentrations of total AXL protein compared to WT as demonstrated by western blot. (**f**) Healthy donor human PB monocyte-derived DCs transfected with miR-34a inhibitor (34ai) showed higher expression levels of AXL compared with cells transfected with control inhibitor (Ci). (**g**) miR-34a[−/−] DCs express higher levels of membrane AXL compared to WT control DCs measured by FACS. WT and miR-34a[−/−] bone marrow DC were cultured in medium (control) or stimulated with LPS (1 ng ml[−1]) for 24 and 48 h and AXL expression evaluated as MFI by cytometry. Data presented as mean ± s.e.m of four biological replicates from two independent experiments. (**h–i**) miR-34a[−/−] had higher levels of *Axl* (**h**) and *Socs-3* (**i**) mRNA in joint tissue compared with WT at day 33 of CIA. Data presented as dot-plots with median lines. *P<0.05, Mann–Whitney *U*-test.

soluble AXL that could function as a decoy receptor for the AXL ligand, GAS6. *Ex vivo* production of pathogenic cytokines, for example, TNF by circulating RA CD1c[+] was reduced by inhibition of endogenous miR-34a expression, whereas mimicking the phenotype of RA DCs by increasing miR-34a in DC from healthy donors enhanced TLRs induced TNF production. *In vivo* miR-34a deficiency led to attenuation of antigen-induced experimental arthritis that was associated with an increased expression of the miR-34a target pathway *Axl/Socs-3* in joint tissue. Consistent with this observation, recent pre-clinical therapeutic studies showed that local and systemic overexpression of GAS6 and Protein S could ameliorate

experimental arthritis[57]. *In vitro* miR-34a[−/−] DCs had constitutively increased levels of AXL and decreased production of pro-inflammatory cytokines (for example, TNF), and Th17 inducing cytokines (for example, IL-6, IL-1β and IL-23) that drive joint pathology. A similar DC phenotype was observed upon active inhibition of endogenous miR-34a in human activated DCs *in vitro*. In addition to the role in regulation of pro-inflammatory mediators, miR-34a supported MHC class II expression, and miR-34a[−/−] DCs had reduced surface MHC class II in immature state and limited upregulation upon maturation. This suggests that miR-34a levels in mature DCs may regulate the signal strength of the MHC/T-cell receptor synapse that controls

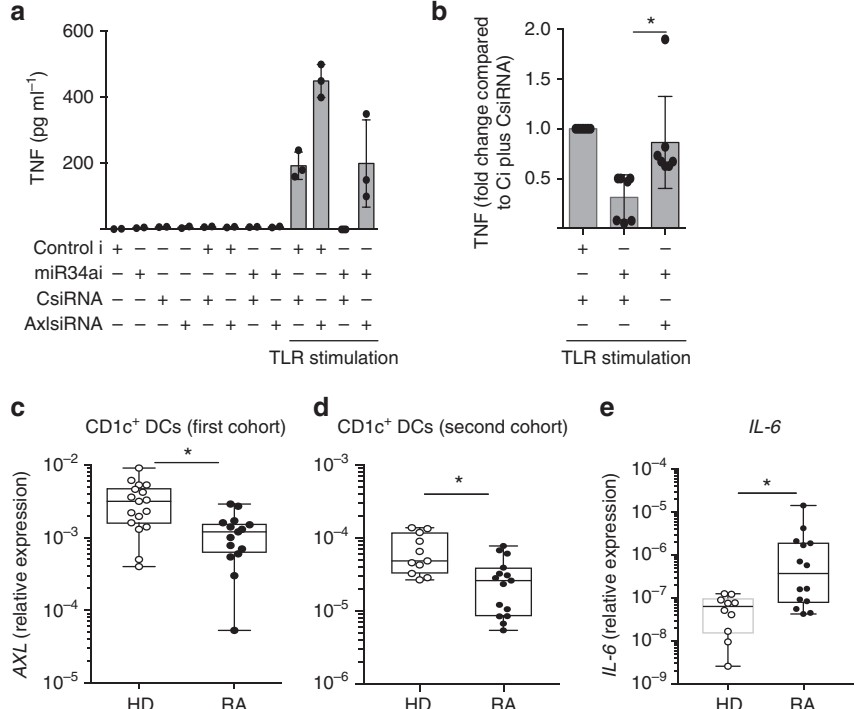

**Figure 6 | AXL expression is constitutively reduced in RA CD1c⁺ DCs.** (a,b) The miR-34a-dependent production of TNF by DCs is mediated by AXL. Healthy donor (n = 7) monocyte-derived DCs were transfected with a combination of miR-34a inhibitor or *AXL* siRNA or control inhibitor or siRNA for 16 h, and then stimulated with CL097 (1 μg ml⁻¹) for 24 h. (a) Representative experimental data for TNF production; mean ± s.d. of 3 biological replicates from one donor. (b) Summary of the relative production of TNF from all 7 donors tested; each normalised to the TNF production by DCs transfected with Control inhibitor (Ci) plus Control siRNA (C siRNA). $P < 0.05$ Kruskal–Wallis with Dunn's multiple comparison test. (c–e) CD1c⁺ DCs in RA patients show decreased expression of *AXL* mRNA and increase expression of IL-6. (c) CD1c⁺ DCs were FACS-sorted from blood of healthy donors (HD, n = 17) and RA patients with established disease (RA, n = 15; cohort 1 described in Supplementary Table 1); and **d**,e CD1c⁺ DCs were FACS-sorted from age-matched healthy donors (n = 11) and RA patients with early and established disease (n = 19, cohort 2 described in Supplementary Table 2). *AXL* (c–d) and *IL-6* (e) mRNAs were evaluated by qPCR. Data are presented as dot-plots with median lines. *$P < 0.05$, Mann–Whitney U-test.

the activation of T cells; in particular Th17 and Th1 which require a strong T-cell receptor signal[58]. This was confirmed by observation of decreased interaction between miR-34a⁻/⁻ DCs and antigen-specific T cells. Commensurate with *ex vivo* and *in vitro* phenotypes upon miR-34a manipulation, deficiency of miR-34a in antigen-presenting cells resulted in the lack of development of antigen-specific Th17 cells, confirmed by low serum concentrations of IL-17; and an attenuated joint pathology in miR-34a⁻/⁻ mice during experimental arthritis.

We observed that miR-34a expression in circulating CD1c⁺ DCs is already increased in some patients with early RA, (≤6 months since diagnosis) as compared to healthy donors, suggesting a role in the onset of disease. The question remains what triggers this increase. The miR-34a promoter is abundant in CpG islands suggesting that environmental factors driving demethylation of the promoter[59,60] is a plausible epigenetic mechanism of regulation as first reported in cancer[59]. In addition, the miR-34a promoter has STAT3-binding sites[54] and a IL-6/STAT3 gene signature was found to be enriched in circulating T cells of early RA patients[61].

In summary, we describe herein that miR-34a controls AXL, a tyrosine kinase receptor that governs the termination of DC activation and contributes to the development of experimental arthritis. In RA patients, sustained expression of high levels of microRNA-34a in CD1c⁺ DCs inhibits cellular AXL expression and this, together with an increased level of decoy receptor soluble AXL in serum, may render DCs more sensitive to maturation signals, which can cause DCs to support auto-reactive T cells (Fig. 7).

Thus, modulating the miR-34a/AXL pathway in RA DCs by using miR-34a inhibitors or AXL agonist could be a feasible therapeutic strategy to restore homoeostasis and promote resolution of RA.

## Methods

**Patients and healthy donors.** For the evaluation of miR-34 and *AXL* mRNA expression in PB, SF and synovial tissue, samples were obtained from healthy donors, RA patients and PsA patients at Rheumatology clinics (Glasgow, UK) and from the Division of Rheumatology, Fondazione Policlinico Universitario A. Gemelli, Catholic University of the Sacred Heart (Rome, Italy). RA patients met the 2010 ACR/EULAR diagnostic criteria[62]. Demographic and clinical information for three cohorts used in the study are detailed in Supplementary Tables 1 and 2. In cohort 1, healthy donors were younger than RA patients, however age was unrelated to miR-34a and *AXL* expression ($r = -0.080$, $P = 0.736$ and r = 0.434, $P = 0.056$, respectively; Pearson correlation coefficient). For the evaluation of GAS6 and sAXL, the concentrations of GAS6 and sAXL were quantified in sera of healthy donors, RA patients and OA patients by commercial enzyme-immunoassay (R&D System, DuoSets: Dy154 and Dy885). Demographic and clinical information are detailed in Supplementary Table 3. The study protocol was approved by the local Ethics Committee of the Catholic University of the Sacred Heart and by the West of Scotland Research Ethical Committee (11/S0704/7). All the donors provided signed informed consent.

**Human DC cultures.** CD14⁺ monocytes from PB were isolated using CD14⁺ micro-beads (#130050201/Miltenyi Biotec) and Auto-MACS separator according to the manufacturer's protocol. Monocyte purity evaluated by flow cytometry was typically >96%. For their differentiation to inflammatory DCs, monocytes were cultured in complete medium (RPMI 1640, 10% fetal calf serum, 2 mM glutamine, 100 U ml⁻¹ penicillin, 100 μg ml⁻¹ streptomycin; Life Technologies) with GM-CSF (100 ng ml⁻¹) and IL-4 (20 ng ml⁻¹; both from Peprotech) for 6-7 days and the DC phenotype was confirmed by their expression of CD11c (#337216/PE/Cy7 anti-human CD11c), CD1c (#331524/APC anti-human CD1c)

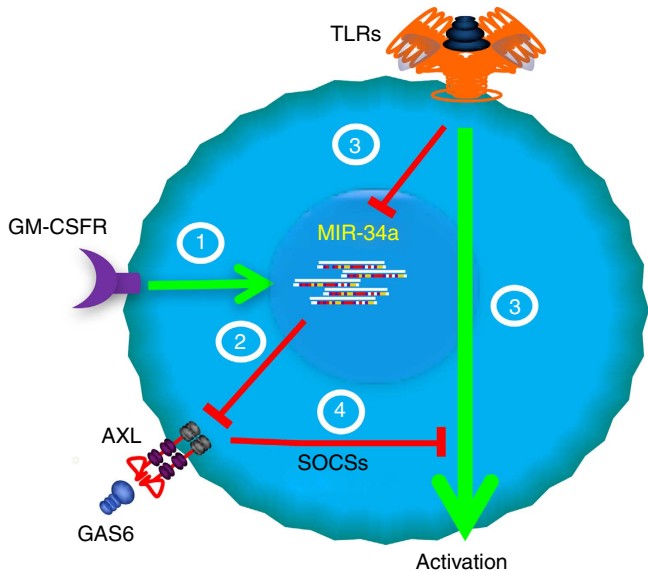

**Figure 7 | A model describing miR-34/AXL function in DCs.** The expression of miR-34a increases during CD1c$^+$ DC differentiation (1) and epigenetically down-regulates AXL expression (2). DC maturation, by TLR ligation, induces DC activation, for example, MHC class II upregulation and cytokine production. This maturation signal also down-regulates miR-34a (3), which then de-represses AXL, which in turn induces SOCSs and terminates DC activation (4), thus providing homeostatic feedback control of DC activation. In RA patients, sustained expression of high levels of miR-34a in CD1c$^+$ DCs inhibits AXL expression and may render DCs more sensitive to maturation signals, which can cause DCs to support sustained auto-reactive T-cell activation.

and MHC class II (#307642/Brilliant Violet 785 anti-human HLA-DR; all antibodies from BioLegend; dilution 1: 100) by cytometry. The conventional CD1c$^+$ population from PB of RA and healthy donors, and from RA and PsA SF and synovial tissue were sorted by FACSAria III according to the sorting strategy presented in Supplementary Figure 1. In preparation for flourescence-activated cell sorting (FACS), SF was overlaid on Histopague-1077 (Sigma) and centrifuged (2,100 r.p.m., 25 min) to obtain mononuclear cells. Synovial tissues were obtained using ultrasound-guided synovial tissue biopsy of the knee following the published protocol[63]. Briefly, all patients underwent ultrasound evaluation of the knee using an ultrasound machine with a multifrequency linear transducer (MyLab Twice, Esaote). Using the ultrasound view, the best point of entrance for the needle was identified on the lateral margin of the suprapatellar recess. Each patient was provided with a face-mask and cap and the whole procedure was undertaken under sterile conditions. Skin disinfection was done with iodine solution (performed twice, starting from the point of needle entrance up to 25 cm proximally and distally). Arthrocentesis of the knee joint was performed using the lateral suprapatellar access if joint effusion was present. The skin, subcutaneous tissue and joint capsule was anaesthetised with 10 ml lidocaine 2%. Next, a 14G needle (Precisa 1410-HS Hospital Service Spa, Italy) was inserted into the joint. Regions of synovial hypertrophy were identified under grey-scale guidance to ensure sampling of representative synovial tissue. All the synovial tissue specimens obtained (at least eight pieces for each analysis) were placed on a nonwoven wet gauze for collection[63].

Synovial biopsies were cut into small pieces and digested with Liberase Research Grade, in RPMI (150 µg ml$^{-1}$ Roche) with rotation in 37 °C for 1.5 h. Digested tissue was passed through a 100 µm tissue-strainer and washed once with PBS/2%FCS/2 mM EDTA. FACS-analysis of single cell suspensions was performed after staining with Fixable Viability Dye eFluor 780 (#65086514/ ThermoFisher Scientific; dilution 1:1,000) to exclude dead cells. Cells were then incubated (4 °C 30 min) with antibody-conjugates, which were from BioLegend and used in 1:100 dilution. These included #307642/Brilliant Violet 785 anti-human HLA-DR, #337216/PE/Cy7 anti-human CD11c, #344104/PE anti-human CD141, #331524/APC anti-human CD1c, #302040/Brilliant Violet 605 anti-human CD16, #304050/Brilliant Violet 711 anti-human CD45, #323004/ fluorescein isothiocyanate (FITC) anti-human CD15 (SSEA-1), #302206/FITC anti-human CD19, #304604/FITC anti-human CD56, #313232/FITC anti-human CD117, #300440/FITC anti-human CD3 and #301822/Alexa Fluor 700 anti-human CD14. The gating strategy was based on fluorescence minus one and unstained controls (Supplementary Fig 1). Briefly, dead and CD56, CD3, CD19, CD15,

CD117 and CD14 lineage-positive cells were excluded, and MHC class II$^+$ CD11c$^+$ cells gated. The CD1c$^+$ population was gated based on the lack of CD141 and CD16 and the presence of CD1c$^+$ expression. CD1c$^+$ cells were then FACS-sorted into tubes with complete media. Blood monocytes were sorted based on their lack of CD2 (#309208/BioLegend; dilution 1:10), CD15 (#323006/BioLegend; dilution 1:10), CD19 (#302208/BioLegend; dilution 1:10), CD56 (#130090755/ Miltenyi Biotec; dilution 1:100) and NKp46 (#331908/BioLegend; dilution 1:20) and the presence of MHC class II (#307633/BioLegend; dilution 1:50). The MHC class II$^+$ cells were then gated based on the expression of CD14 (#325608/ BioLegend; dilution 1:20) and CD16 (#302006/BioLegend; dilution 1:20); resulting in three monocyte populations CD14$^+$CD16$^-$; CD14$^+$CD16$^+$, CD14$^{dim}$CD16$^+$. Sorted DCs and monocytes were used for cell culture experiments or were immediately lysed with Qiagen buffer and RNA isolated with micro-miRNAeasy kit (#217084) or miRNeasy mini Kit (#217004) according to the manufacturer's protocol (Qiagen). Cultured DCs in complete medium were transfected either with miR-34a inhibitor or control inhibitor (both at 20 nM; Qiagen) or with miR-34a mimic or control mimic (*C. elegans* miR-67), both at 20 nM (Dharmacon) using DharmaFECT 3 transfection reagent (ThermoFisher Scientific). For some experiments, DCs were transfected with AXL-specific siRNA or All Stars Negative Control siRNA (40 nM, both from Qiagen). To monitor transfection efficiency DCs were transfected with Dy547-labelled mimic or All Stars Negative Control siRNA labelled with Alexa Fluor 488. After 16 h, cells were stimulated with LPS (1 ng ml$^{-1}$) or CL097 (1 µg ml$^{-1}$, both from InvivoGen) and cells and supernatant were collected after 24 h. In all experiments, the transfection efficiency of siRNA and microRNA mimic was checked by flow cytometry and only when >60% of cells were positive were they used for further analysis.

**Mice.** MicroRNA-34a gene-deficient (miR-34a$^{-/-}$) mice on a C57BL/6 background were purchased from Jackson Laboratories[64]. WT littermates were obtained by backcrossing miR-34a$^{-/-}$ mice with C67BL/6 mice (Jackson Labs). Axl$^{-/-}$ mice and WT littermates C57BL/6 (B6Nq; Johan Bäcklund; Karolinska Institute, Sweden) were bred and maintained in a pathogen-free facility at University of Glasgow. Procedures were approved by University of Glasgow Ethical Committee and the UK Home Office (project licence 60/4430). *In vivo* experiments were conducted on males age 8–12 weeks. *In vitro* experiments were conducted on cells from males and females age 8–12 weeks.

**Collagen-induced arthritis.** To induce collagen-induced arthritis, mice were injected intradermally with type II chicken collagen emulsified in Freund's complete adjuvant (200 µg in 0.2 ml. MD BioSciences) on day 0. On day 21, type II chicken collagen/PBS (200 µg in 0.2 ml) was injected i.p[65]. Mice were monitored for development of arthritis by microcaliper measurements of paw swelling. In addition, a clinical severity score was used in which 0 = no reaction; 1 = mild but definite redness and swelling of the ankle/wrist/digits; 2 = moderate-to-severe redness and swelling of the ankle/wrist; 3 = redness and swelling of the entire paw including digits; and 4 = maximally inflamed limb with involvement of multiple joints[66]. Mice were killed on day 33 and cells and tissues harvested. Two researchers measured paw swelling (one of them blind); and both evaluated histology sections blind to experimental condition.

*Adoptive transfer of OT-II T cells.* OT-II T cells on a CD45.1 background (at 0.5 and 1.5 × 10$^6$) were injected into the tail vain of WT and miR-34$^{-/-}$ mice on day 0. On day 1, mice were injected with 100 µg of OVA peptide or PBS per mouse (Cambridge Reserved Biochemical) in complete Freunds adjuvant (0.5 mg ml$^{-1}$; Sigma) in 0.2 ml into the right leg muscle. On day 6, mice were culled, the draining lymph nodes harvested and dispersed cells re-stimulated with PMA/ionomycin (50/500ng ml$^{-1}$; Sigma) overnight in the presence of Golgi Stop (Life Technologies), and analysed for the intracellular production of IL-17, IFN-γ and IL-5 using fluorescently labelled antibodies (#559502/PE anti-mouse IL-17A; BD Pharmingen; #505810/APC anti-mouse IFNγ; BioLegend; #504311/BV421 anti-mouse IL-5; BioLegend; all in 1:50 dilution and Fix/Perm kit according to protocols (#420801 and # 421002, BioLegend).

**Mouse DC culture.** Bone marrow cells from WT and miR-34a$^{-/-}$ mice were cultured with GM-CSF (20 ng ml$^{-1}$, PeproTech) at a concentration of 0.2 × 10$^6$ ml$^{-1}$ in 10 ml of complete medium in petri dishes. After 3 and 6 days the medium was replaced with fresh medium supplemented with GM-CSF. On day 8, cells were seeded into 24-well culture plates at a concentration of 0.25 × 10$^6$ ml$^{-1}$ and stimulated with LPS (1 ng ml$^{-1}$) or CL097 (1 µg ml$^{-1}$). Cells and culture supernatant were collected at various time points for analysis of cytokine production, cell surface markers and AXL protein. For some *in vitro* experiments CD11b$^+$CD4$^+$ spleen cells were isolated with CD4$^+$ DC isolation kit (#130091262/Miltenyi Biotec).

**Evaluation of DC/T cell interaction with IN Cell Analyzer 2000.** Spleens were isolated from OTII transgenic mice and prepared into single cell suspensions and treated with 5 ml of 1 × red blood cell lysis buffer (eBioscience) for 5 min at RT. CD4$^+$ T cells were isolated from splenocytes by negative selection (#130095248/ Miltenyi Biotec). Bone marrow derived WT and miR-34a$^{-/-}$ DCs and CD4$^+$ T cells were re-suspended at a density of 10$^7$ cells per ml in serum-free RPMI-1640

medium and labelled with Cell Tracker Red CMTPX or CFSE cell tracer kits (#C34552 and #C34554 respectively; Molecular probes) at 7.5 μM, respectively, for 10 min in a 37 °C incubator. After incubation, cells were washed by centrifugation at $300 \times g$ for 5 min at 4 °C three times in RPMI medium containing 10% FCS to deactivate the dye. DCs and CD4$^+$ T cells were co-cultured in 384-well μ-clear tissue culture treated micro-plates (Greiner) at ratios of 1:1 (8.000 of each cell type) or 1:3 with ovalbumin peptide (OVA) at 1 μg ml$^{-1}$; with or without lipopolysaccharide (LPS) at 100 ng ml$^{-1}$ for 24 h.

**Image acquisition.** IN Cell Analyzer 2000 is a lamp-based high content analysis system equipped with a large chip CCD camera (Photometrics CoolSNAP K4) allowing for whole-well imaging. The Nikon × 10 magnifying objective with flat field and apochromatic corrections (plan Apo), chromatic aberration-free infinity (CF160) and 0.45 numerical aperture (NA) was used. Cells were imaged at 37 °C and 2D images of CFSE-labelled CD4$^+$ T cells were acquired in the FITC channel using 490/ × 20 excitation, 525/36 m emission. CMTPX-labelled DCs in the TexasRed (TR) channel were acquired using 555/ × 25 excitation and 605/52 m emission. One tile was imaged per well covering ∼20% of the well area. Images acquired by the IN Cell Analyzer 2000 were analysed using the Developer Toolbox software V1.9.2 (GE Healthcare). Data are shown as representative screenshots of a 384-well or the percentage of OTII T cells that overlap with DCs.

**Western blot for AXL.** Bone marrow derived DCs were seeded in complete media into six-well plates ($1 \times 10^6$ per well). Human PB monocyte-derived DCs ($1 \times 10^6$ per well) were transfected with control or miR-34a inhibitor (both 50 nM, Qiagen). After 24 h (mouse DCs) or 48 h (human DCs) whole-cell lysates were prepared using M-PER (ThermoFisher Scientific) lysis buffer. Lysate samples (10 μg protein) were run on a 4–12% SDS–polyacrylamide gel electrophoresis gel, followed by transfer to polyvinylidene difluoride membrane and incubation with either goat anti-mouse AXL (1:200 dilution, #/AF854/R&D Systems), rabbit anti-human AXL (1:500 dilution, #8661S/Cell Signaling Technology), or anti-mouse/human β-actin control (1 μg ml$^{-1}$, #3c-81178/Santa Cruz Biotechnology).

**Flow cytometry of DC surface markers.** Expression of DC surface markers was evaluated by LSR-II cytometer (BD Bioscience) using the following antibodies: anti-MHC class II-PE-Cy7 (#107629/BioLegend) or APC-Cy7 (#107628/BioLegend) anti-CD11c-FITC (#117306/BioLegend) or PerCP-Cy5.5 (#560584/BD Pharmingen), anti-CD86-PerCP (#105025/BioLegend), anti-CD40-Pacific Blue (#124626/BioLegend), anti-PD-DL-1-Qdot-605 (#124321/BioLegend), ICAM-1-FITC (#11054182/eBioscience), CD18-PE (#101408/BioLegend), CD69-PE-Cy7 (#25069182/eBioscience), CD11a-APC (#101120/BioLegend), CD44-e450 (#48044182/eBioscience), CD11b BV605 (#83011242/eBioscience; all used in 1:100 dilution), and anti-AXL-APC (#FAB8541A/R&D Systems; dilution 1:20). The dead cells were excluded with Fixable Viability Dye eFluor 780 or e506 (#650866/ThermoFisher Scientific). There was no difference in viable cell proportion of BM-derived DCs between WT and miR-34a$^{-/-}$ genotypes (mean ± s.d: 75% ± 12 and 71% ± 10, respectively).

**AXL 3′-UTR Luciferase assay.** Human AXL 3′-UTR in sense or anti-sense orientation were cloned downstream of the Renilla Luciferase gene in pmiRGLO vector (Promega). HEK293 cells were co-transfected with 0.2 μg pmiRGLO containing sense or anti-sense AXL 3′-UTR and 40 nM miR-34a mimic or scrambled mimic control, using Attractene (Qiagen). Luciferase activity was measured 24 h later using the Dual-GLO Luciferase assay system (#E2920/Promega)[67].

**Quantitative PCR.** Total DC RNA was extracted using a miRNeasy mini Kit (#217004) or micro-miRNAeasy kit (#217084) (both from Qiagen). For microRNA analysis, cDNA was transcribed from RNA using either a miScript Reverse Transcription Kit II (#218160 Qiagen) or TaqMan microRNA Reverse Transcription Kit (#4366596/Life Technologies). For mRNA analysis, a High Capacity cDNA Reverse Transcription Kit was used (#4368814/ThermoFisher Scientific). TaqMan mRNA or miRNA assays (Life technologies) or miScript primer assays (Qiagen) were used for semi-quantitative determination of the expression of miR-34a (Hs000426 Life Technologies; MS00001428 Qiagen), miR-34c (Hs000428), Let7a (Hs000377, U6B snRNA (Hs001973; MS00033740), TNF Mm00443258_m1), AXL (Hs01064444_m1, Mm00437221_m1), Socs3 (Mm00545913_s1), p19 (Mm00518984_m1), IL-6 (Hs00174131_m1) and 18S (Hs03003631_g1) with miRScript SybR Green PCR kit (#218073/Qiagen) or TaqMan Gene Expression master mixes (Life Technologies). In some experiments miR-34a copy number was evaluated. Known copy numbers of synthetic miR-34a mimic and housekeeping let7a (both from Dharmacon) were used to prepare a standard curve of serial 10-fold dilutions in nuclease-free water (range from $1 \times 10^9$ to $1 \times 10^3$ copies). Each of these standards was used alongside cDNA from the samples in qPCR (miScript Sybr Green PCR kit) with specific human miR-34a and let7a primers (Qiagen) on a 7900HT TaqMan reader. Based on the standard curves, the copy number of miR-34a and let7a the samples were calculated and data were presented as the copy number of miR-34a per 10,000 copies let7a. In some

experiments, the expression of miR-34a is presented as a relative value (i) $2^{-\Delta CT}$, where $\Delta Ct = $ Cycle threshold for RNU6 (housekeeping) minus Ct for miR-34a or (ii) fold change, where $\Delta Ct$ for selected control condition = 1 or 100%.

To compare miR-34a and AXL expression in synovial CD1c$^+$ DCs with blood CD1c$^+$ DCs, amplification of cDNA was necessary due to the small number of CD1c$^+$ DCs obtained from synovial biopsies (mean cell number: 2,000 cells) by FACS-sorting. Equal numbers of blood, SF and synovial tissue CD1c$^+$ DCs were used and RNA isolated (cohort 2; Supplementary Table 2). RNA was divided into two aliquots for separate transcription into cDNA with either High Capacity Reverse Transcription kit (#4368814/ThermoFisher Scientific) to evaluate AXL expression or with miScript Reverse Transcription Kit II (#218160/Qiagen) to evaluate miR-34a expression. In the next step, cDNA for AXL, IL-6 and housekeeping 18S were amplified with PreAmp Master kit (#4391128/ThermoFisher Scientific) and cDNA for RNU6, miR-34a were amplified with miScript PreAMP PCR Kit (#331452/Qiagen) followed by standard qPCR as described above.

**Luminex and enzyme immuno-assays.** The concentration of cytokines and chemokines in mouse and human DC culture supernatants was quantified by immuno-fluorescence assay using a 20-plex cytokine assay (LMC0006; Invitrogen) on a Bio-Plex platform (Bio-Rad) or using paired antibodies ELISAs (mouse and human TNF, #CHC3013 and #CHC1753/Life Technologies; mouse IL-23 (#88723088/ ThermoFisher Scientific) and mouse GAS6 (#DY986/R&D Systems)[66].

**Statistical analysis.** The data were analysed by Graph Pad Prism version 5.0 and Minitab 17 software. Data was summarized as mean ± s.d./s.e.m. or median ± IQR, and between-group differences were tested by t-test, by paired t-test or by Mann–Whitney u-test, or Kolmogorov–Smirnov test, depending on data distribution, or by ANOVA or Kruskal–Wallis tests with correction for multiple comparisons. Correlation between normally distributed continuous variables was tested using the two-tailed Pearson Correlation Coefficient. A P-value <0.05 was considered significant.

**Data availability.** The data that support the findings of this study are available from the corresponding author upon request.

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

## Acknowledgements

We thank the RA patients and healthy donors who participated in this study. We would thank Mrs Diane Vaughan for her outstanding support in cell sorting and phenotyping; Dr Eric Kalkman for his help with IN Cell Analyzer 2000; Dr Tom Barr for his help with adoptive transfer model; and Drs Siebert and Zhao for some of the PsA samples. We thank Mr David Riggans from University of Glasgow Mail Room who facilitated smooth deliveries of clinical samples from Rome to Glasgow. MKS was supported by Arthritis Research UK (ARUK) Career Development Fellowship (♯19213). SA was supported by a fellowship from Novo Nordisk UK. CT was supported by the Oliver Bird Rheumatism Programme/ Nuffield Foundation. EGM was supported by European League Against Rheumatism (EULAR) and the Catalonian Society of Rheumatology. VM was supported by ARUK Fellowship (♯20848). This work was performed in, and support by, the ARUK Centre of Excellence for the Pathogenesis of Rheumatoid Arthritis (RACE/♯20298).

## Author contributions

M.K.-S. designed and performed experiments, analysed data and wrote the manuscript. S.A. contributed to designing the study, performing experiments, analysing the data, and contributed with clinical samples; E.G.M., C.T., Ch.K. and L.S., G.D.S. and C.D.M. performed experiments. D.S.G. and V.M., provided the technology and performed

experiments. A.Z. performed experiments and analysed clinical data. B.T. and L.P. provided clinical samples and performed experiments. D.P., R.D.B., D.M. provided clinical samples and analysed clinical data. J.M.B., G.L., S.M. and E.G. provided technologies. C.M. and G.F. contributed to designing the study and writing the manuscript. I.B.M. contributed to designing the study, analysing the data and writing the manuscript.

## Additional information

**Competing interests:** The authors declare no competing financial interests.

