## [Peer Review File · Nature Communications]

Reviewers' comments:

Reviewer #1 (Remarks to the Author):

In this manuscript, Kurowska-Stolarska et al analyze the functional role of miR-34a, which they found by transcriptomic analysis, to be elevated in dendritic cells (DCs) from rheumatoid arthritis (RA) patient PB and SF. They demonstrate its target axl is regulated by miR-34a and that knocking out miR-34a reduces inflammation and differentiation of Th17 cells in collagen-induced arthritis. miR-34a-deficient DCs secrete lower levels of pro-inflammatory cytokines and express lower levels of MHC class II but not costimulatory molecules. They also cluster less strongly with T cells in vitro. In patients, regardless of disease activity, the authors find elevated gas6 and sAXL (decoy receptor for axl). Strengths: the work on miR-34a in DCs and demonstrating a potential role in rheumatoid arthritis is novel, interesting and informative for the field. The experiments are well done, with adequate numbers of replicates, statistical analysis and effective use of alternate strategies to confirm their findings. Weakness: the strong thread that links the data at the beginning gets confusing by the end of the paper. This could easily be improved with revision.

Specific points

1. Given the lower DC-T cell interaction by imaging, was there a reduction in adhesion molecule expression in the DCs knocked out for miR-34a? Is there an explanation for this reduced interaction?
2. In view of the increased Th17 differentiation in mice with wt miR-34a, was there a difference in p19 (IL-23) expression in wt mice as compared to miR-34a deficient mice?
3. Figure 6 and related text is quite confusing. Some of the explanation regarding the decoy receptor comes when fig 6 is described, some in figure 7 and some in the discussion. I found it difficult to put all the pieces together in light of figs 6D and E. There is no correlation of gas6 or sAXL levels with RA treatment response, leading me to wonder about the relevance of this system. Is there a correlation of DC activation/cytokine production with DMARD treatment response in patients?

Reviewer #2 (Remarks to the Author):

The authors have investigated the role of miR-34a in regulation of inflammatory response activity of dendritic cells (DC). They demonstrate that miR-34a is upregulated in DC from peripheral blood and is high in DC from synovial fluid - even though in this case no control DC were available. The expression of mir-34a was strongly upregulated during maturation of monocytes into DC and subsequently down regulated by TLR agonist or LPS treatment. MiR-34a^{-/-} mice were found to be resistant to experimental arthritis and demonstrated lower IL-17 in circulation - thought to be due to decreased number of IL-17 producing cells. Mir-34a is further demonstrated to drive DC activation and DC from the knock-out mice produce less pro-inflammatory cytokines. Mir-34a is known to down regulate a large number of genes including the Axl receptor and the authors decided to focus on Axl as it has been shown to be involved in immune-cell regulation. The authors confirm that Axl is a target for mir-34a in DC and moreover that Axl expression is increased in mir-34a^{-/-} mice. Axl expression is also found to be decreased in DC from RA patients. The findings related to the importance of miR-34a in regulation of DC function are novel, whereas the connection between mir-34a and Axl are well known - also the importance of Axl in immune-cell regulation is well documented previously. Overall I find the study very interesting and worthwhile. The data from mouse experiments are convincing and well documented. The experimental results related to the human RA patients are less convincing partly due to the small cohort studied but also to lack of a convincing connection/correlation between the increased mir-34a in DC and the decreased mRNA expression of Axl in DC from RA patients. The authors need to address the following concerns:

- 1, The patient cohort is small - in figure 1, only 6 patients and 6 controls are compared in A and in B, the controls are not matched in number to studied patients. The DC from the synovial fluid of RA patients are not compared with SF from other types of inflammatory knee diseases.
- 2, In figure 6, Axl mRNA is measured in 15 patients with RA and in controls. The controls and patients were not matched, which together with the small number makes the data less convincing- was there a correlation between the Axl mRNA level and that of mir-34a in these DC? Was the Axl protein level lower in DC from RA as compared to controls?
- 3, It seems something is wrong with the Gas6-measurements in Fig 6D, as a majority of measured controls and patients have undetectable Gas6-levels. Several previous studies have shown that the normal plasma concentration of Gas6 lies just below 20 ng/ml.
- 4, In figure 6E, the differences between controls and patients is very small (mainly due to a few outliers) and hardly justifies the suggestion in the last sentence of the results that the increased sAxl would prevent engagement of Gas6-Axl inhibitory pathway in RA DC.
- 5, Are the significance indications in Fig 6 D and E the same for all compared groups? Is this a nonparametric test or a Kruskal-Wallis test?
- 6, In Supplemental Fig 1, the sorting strategy of the cells is shown. It is not clear, however, why the CD16-CD141- population is defined as CD1c+ per default.
- 7, Many figures lack indications for statistical significance. If the data is analyzed but not significant, please indicate so in the figure, especially in figures where it is not evident that it is not (e.g. Fig2C or 3H)
- 8, Are the stimulations in figure 2 C-D done on the whole monocyte population or on a subpopulation?
- 9, Why do the western blots measuring Axl expression look so different in figure 5 E and F?
- 10, Is Gas6-production different between miR-34a-/- and wt CD1c+DCs? cells? Is there more Gas6 on the surface of miR-34a-/- CD1c+ DCs compared to wt DCs?
- 11, To confirm that the miR-34a regulated effects on cytokine production and antigen presentation are opposed in Axl KO mice, the authors should perform similar experiments as in Fig 4 on CD1c+ DCs from Axl KO and wt mice, especially regarding MHC surface expression, DC-T-cell interactions upon OVA challenge and cytokine production.

Minor points:

- 1, Page 3, line 7: woman>women
- 2, Page 12, line 16: Sentence should start "In RA patients..."

Reviewer #3 (Remarks to the Author):

The work is original, and propose identification of a new pathway (miR34a/Axl/Gas6) to understand DC activation and autoimmunity. First , the author identify miR34a as increased in blood CD1c+ DCs from RA patients than

healthy control subjects, and this was further increased in SF DCs. Moreover miR34 is induced by GM-CSF. These points raise some questions : Does inflammatory status or use of anti TNF change expression ? is this specific to RA or find similar results in other chronic inflammatory diseases ?

In a second step, the authors provide animal experiments and convincing data in miR-34a-/- mice had a reduced incidence and severity of arthritis. However, it is not clear if this effect is dependent only on miR34 expression in DC or if miR34a has other target in immune cells or synovial cells. They show a decrease in Th17 number, is it direct effect on T cells or indirect on DC? but what is the impact on T cells functions ? B cells in miR34-/- mice?

pathway: they report that microRNA-34a has 416 putative targets, among others bcl2 , caspase... Did the author check apoptosis pathway of DC before focusing on Axl? pro-inflammatory cytokines (TNF α , IL6) was reduced by ex vivo gene-silencing of miR-34a which was restored by down-regulation of AXL suggesting that the control of DC activation is via regulation of tyrosine kinase receptor AXL. At last, RA patients had higher concentrations of GAS6 and sAXL in the serum

independently of their disease activity, but the distribution seems very large in different group of patients, and in methods patient treatment seems heterogeneous. Again, checking in other inflammatory diseases to assess specificity would be relevant. define subgroup of patients according to sAXL level may provide information? i sth elevel GAS6/sAxl changing overtime?

Point-by-point replies to reviewer comments

Reviewer #1:

In this manuscript, Kurowska-Stolarska et al analyze the functional role of miR-34a, which they found by transcriptomic analysis, to be elevated in dendritic cells (DCs) from rheumatoid arthritis (RA) patient PB and SF. They demonstrate its target axl is regulated by miR-34a and that knocking out miR-34a reduces inflammation and differentiation of Th17 cells in collagen-induced arthritis. miR-34a-deficient DCs secrete lower levels of pro-inflammatory cytokines and express lower levels of MHC class II but not costimulatory molecules. They also cluster less strongly with T cells in vitro. In patients, regardless of disease activity, the authors find elevated gas6 and sAXL (decoy receptor for axl). Strengths: the work on miR-34a in DCs and demonstrating a potential role in rheumatoid arthritis is novel, interesting and informative for the field. The experiments are well done, with adequate numbers of replicates, statistical analysis and effective use of alternate strategies to confirm their findings.

We appreciate this recognition of our concept, our efforts and the potential value of our findings.

Weakness: the strong thread that links the data at the beginning gets confusing by the end of the paper. This could easily be improved with revision.

We acknowledge this observation. It was due to the limited number of well-defined clinical samples for the analysis of microRNA34a/Axl in arthritis patients. We addressed this by (i) substantially increasing the number of RA patient blood CD1c⁺ analysed and (ii) adding evaluation of synovial tissue CD1c⁺; (iii) increasing the number of RA CD1c⁺ in functional studies; and (iv) by investigating the levels of soluble AXL and GAS6 in additional RA cohort and in other arthropathies.

Together, these additional experiments confirmed our initial findings that microRNA34a/Axl pathway is deregulation in RA DCs and that this contributes to their function. We also confirmed increased concentrations of soluble AXL (an inhibitor of Axl pathway) in the serum of an additional cohort of RA patients and revealed that this increase was most profound in early, naïve RA patients. These additional findings allowed us to revise the latter part of the paper with more confidence and with greater clarity.

We addressed the specific reviewer's queries point-by-point below

Specific points

1. *Given the lower DC-T cell interaction by imaging, was there a reduction in adhesion molecule expression in the DCs knocked out for miR-34a? Is there an explanation for this reduced interaction?*

We found no difference in the expression of the major DC adhesion molecule involved in synapse formation with T cells, ICAM-1, and the key DC integrins CD11b, CD11c and CD18, between WT and miR34a^{-/-} DCs by flow cytometry. We speculate that decreased levels of MHC class II (shown in Figure 4B-C) may contribute sufficiently to the reduce interactions between miR34a^{-/-} DC and antigen specific T cells as compared to WT DCs. Another possible explanation is that, as AXL signalling is known to regulate the actin cytoskeleton (Thuraia et al., Moll Cell Biol., 2015, 35, 76) alterations in the DC cytoskeleton may influence contact dynamics with T cells in miR34a^{-/-} DC (Comrie at al, J.Cell.Biol., 2015, 208:457).

This information is added in Supplementary Figure 5 and included in the text (page 8, lines 2-4).

2. *In view of the increased Th17 differentiation in mice with wt miR-34a, was there a difference in p19 (IL-23) expression in wt mice as compared to miR-34a deficient mice?*

In the ankle joints of WT and miR34a^{-/-} mice with collagen-induced arthritis we found reduced expression of p19 mRNA in miR34a^{-/-} mice as compared to WT controls. We also measured the concentrations of IL-23 produced by WT and miR34a^{-/-} bone marrow derived DCs in culture by enzyme immuno-assay. While IL-23 was found in supernatants from WT DCs stimulated with LPS, it was undetectable in supernatants from miR-34a^{-/-} DCs. This suggests that miR-34a supports the production of IL-23 by DCs. This information is now provided in the text (page 7, line 25), and presented on Supplementary Figure 4C.

3. *There is no correlation of gas6 or sAXL levels with RA treatment response, leading me to wonder about the relevance of this system*

We sought to clarify the relationship between treatment response and soluble AXL (sAXL) in a prospective study by collecting an addition validation cohort of serum samples from RA patients (Cohort 3). The cohort demography is presented in Supplementary Table 3, and included:

- early RA patients that were naïve to DMARDs treatment (n=20);
- RA patients that were resistant to DMARDs or a combination of DMARDs and TNF inhibitors treatment (n=20);
- RA patients in a stable remission induced by a combination of Methotrexate and TNF inhibitor treatment (n=10);

- healthy donors (n=10)

We also included non-inflammatory osteoarthritis (OA) patients (n=10) as a control for joint disease that does not have an autoimmune component.

Soluble AXL was significantly upregulated in a naïve to treatment and resistant to treatment compared to healthy; with the highest concentration of sAXL detected in sera of a naïve to DMARDs group: median 29.02 (interquartile range: 26.27-33.33) compared to 16.2 (interquartile range 13.4 - 20.4) of the healthy donors; ($p=0.0007$). The RA in remission group showed upregulated levels of sAXL compared to healthy donors; however the levels were quite variable between the patients and it did not reach statistical significance. The serum concentrations of sAXL in OA were similar to that of healthy donors and significantly lower than RA patients ($p=0.029$). The lack of a clear downregulation of sAXL in RA patients in remission and low levels of sAXL in OA suggest that elevated levels of serum sAXL in RA could be associated with underlying autoimmunity per se and to a lesser extent to disease activity measures (inflammation). We plan to investigate this in the future by testing sAXL levels in ACPA positive pre-clinical RA and in sequential blood samples of patients in remission upon treatment withdrawal. We suggest that this is a significant study in its own right and goes beyond the present manuscript scope. We included these new data in Supplementary Figure 8A and in the text (pages 10-11, lines 27 and 1-12)

We quantified serum concentration of GAS6 in this prospective cohort and found no significant difference between groups. The data from the new cohort are presented in Supplementary Figure 8B and included in the text (pages 10-11, lines 27 and 1-12).

Is there a correlation of DC activation/cytokine production with DMARD treatment response in patients?

In order to investigate the relationship between treatment and activation of DC we evaluated IL-6 expression in FACS-sorted blood CD1c⁺ dendritic cells of RA patients (n=19; Cohort 2) who were either naïve-to-treatment (n=7) or resistant-to-treatment (methotrexate n=9; TNF inhibitor n=3), both groups with high disease activity; and age-matched healthy donors. The clinical and demographic information for this cohort of patients is presented in Supplementary Table 2.

IL-6 expression was increased in circulating CD1c⁺ DCs of the RA patients (new Figure 6E) compared to healthy control subjects (n=10). There was no statistically significant difference in IL-6 expression between naïve and treatment-resistant RA. The IL-6 expression negatively correlated with the AXL mRNA expression in these cells ($r = -0.56$; $p=0.04$) confirming the negative relationship between pro-inflammatory cytokine production and AXL expression shown in functional studies (Figure 6.A-B).

These data suggest that high disease activity is associated with the presence of activated DCs in circulation. The new data is presented on Figure 6E, and discussed in the text on page 10, line 18-21.

Figure 6 and related text is quite confusing. Some of the explanation regarding the decoy receptor comes when fig 6 is described, some in figure 7 and some in the discussion. I found it difficult to put all the pieces together in light of figs 6D and E.

To help resolve the confusion, these figures have been updated with newly generated data and are now presented as Supplementary Figure 8. These new data are explained in the result section on pages 11 (line 27) and 12 (lines 1-27) and the pathway is summarised in the discussion on pages 14 (lines 26-27) and 15 (lines 1-5).

In the summary, we propose that in RA patients, sustained expression of high levels of microRNA34a in CD1c⁺ DCs inhibits cellular AXL expression and this together with an increased level of decoy receptor soluble AXL in the circulation may render DCs more sensitive to maturation signals, which can cause DCs to support auto-reactive T cells.

Reviewer #2:

The authors have investigated the role of miR-34a in regulation of inflammatory response activity of dendritic cells (DC). They demonstrate that miR-34a is upregulated in DC from peripheral blood and is high in DC from synovial fluid - even though in this case no control DC were available. The expression of mir-34a was strongly upregulated during maturation of monocytes into DC and subsequently down regulated by TLR agonist or LPS treatment. MiR-34a^{-/-} mice were found to be resistant to experimental arthritis and demonstrated lower IL-17 in circulation - thought to be due to decreased number of IL-17 producing cells. Mir-34a is further demonstrated to drive DC activation and DC from the knock-out mice produce less pro-inflammatory cytokines. Mir-34a is known to down regulate a large number of genes including the Axl receptor and the authors decided to focus on Axl as it has been shown to be involved in immune-cell regulation. The authors confirm that Axl is a target for mir-34a in DC and moreover that Axl expression is increased in mir-34a^{-/-} mice. Axl expression is also found to be decreased in DC from RA patients. The findings related to the importance of miR-34a in regulation of DC function are novel, whereas the connection between mir-34a and Axl are well known - also the importance of Axl in immune-cell regulation is well documented previously. Overall I find the study very interesting and worthwhile. The data from mouse experiments are convincing and well documented.

We appreciate the recognition of our concept, our efforts and the potential value of our *in vivo* work

The experimental results related to the human RA patients are less convincing partly due to the small cohort studied but also to lack of a convincing connection/correlation between the

increased mir-34a in DC and the decreased mRNA expression of Axl in DC from RA patients.

We agree with the reviewer and addressed that by (i) substantially increasing the number of RA patient blood CD1c⁺ analysed and (ii) adding evaluation of synovial tissue CD1c⁺; (iii) increasing the number of RA CD1c⁺ in functional studies with miR34a inhibitor (n=7); (iv) investigating the levels of soluble AXL and GAS6 in additional RA cohort and in other arthropathies; and (v) by investigating the correlation between miR-34a and AXL expression in RA patients.

We addressed the specific reviewer's queries point-by-point below

1. The patient cohort is small - in figure 1, only 6 patients and 6 controls are compared in A and in B, the controls are not matched in number to studied patients. The DC from the synovial fluid of RA patients are not compared with SF from other types of inflammatory knee diseases.

To answer this question, firstly we increased the RA patient and healthy donor numbers. We set up a prospective study of blood and synovial tissue dendritic cells from RA patients. We FACS-sorted CD1c⁺ from RA peripheral blood (PB; n=19), from RA synovial tissue (n=9) and age-matched healthy donors (n=10). Secondly, we extended the study to include dendritic cells from Psoriatic Arthritis blood (n=16) and synovial tissue/fluid (n=5). The clinical and demographic information is presented in Supplementary Table 2.

This second cohort confirmed the findings in the first (original) study cohort of an increased expression of miR-34a in PB CD1c⁺ dendritic cells of RA compared to HD, and revealed a further increase in miR-34a expression in synovial tissue CD1c⁺ dendritic cells (new Figure 1C). PsA PB CD1c⁺ dendritic cells did not show significant differences in miR-34a expression compared to PB cells of healthy donors. We observed an increase in miR-34a in PsA synovial tissue/fluid CD1c⁺ dendritic cells compared with healthy circulating CD1c⁺ (new supplementary Figure 2); however the upregulation was significantly lower than in RA synovial tissue CD1c⁺ (p=0.02). In summary, these data confirmed a potential strong contribution of miR-34a to the dendritic cells function within the structure of RA synovial tissue.

Our data also suggest a potential involvement of miR-34a in driving immune response in PsA joints however to a lesser degree than in RA. These points are discussed in the text on pages 6 (lines 19-27) and 7 (lines 1-8).

2. In figure 6, Axl mRNA is measured in 15 patients with RA and in controls. The controls and patients were not matched, which together with the small number makes the data less convincing- was there a correlation between the Axl mRNA level and that of mir-34a in these DC?

We agree with the reviewer and increase the number of investigated clinical samples (n=19) and age-matched controls (n=10) and evaluated *AXL* mRNA expression in this newly collected cohort (described in point 1; cohort 2, Supplementary Table 2). These new samples confirmed decrease in *AXL* mRNA expression in CD1c⁺ dendritic cells of RA patients compared to healthy controls that was associated with the upregulation of miR-34a that correlated negatively with *AXL* (r= -0.463, p=0.04). We present these new data on Figure 6D and included this new information into text page 11 (lines 18-21).

Was the Axl protein level lower in DC from RA as compared to controls?

While human and mouse *AXL* protein is convincingly detectable by Western blot (as shown on Figure 5E-F), and mouse *Axl* by flow cytometry (Fig 5G), we could find no good antibody or strategy to detect human *AXL* by flow cytometry and the low cell patients number precluded analysis by Western blot. Therefore, we were unable to provide a systematic characterisation of *AXL* protein expression on DCs. However, we believe that the *AXL* mRNA expression data in the new cohort of patients, *AXL* western blot and functional experiments with *AXL* siRNA support the role of miR-34a/*Axl* in RA DC activation.

3. It seems something is wrong with the Gas6-measurements in Fig 6D, as a majority of measured controls and patients have undetectable Gas6-levels. Several previous studies have shown that the normal plasma concentration of Gas6 lies just below 20 ng/ml.

We appreciate this comment and are grateful for the opportunity to revisit this assay. We measured *GAS6* and s*Axl* in serum diluted appropriately for s*Axl* but, guided by your comment, this was not suitable for *GAS6*. We repeated the *GAS6* ELISA assay on the serum collected from an additional RA study groups; and included serum from osteoarthritis (OA) patients and healthy control serum samples (Cohort 3, Supplementary Table 3). The concentration of *GAS6* in the additional healthy control subjects were consistent with other studies; median 8.7ng/ml (interquartile range: 0.125-10.5) and we found no significant difference between any of the patients' groups and control samples (Supplementary. Fig 8B).

4. In figure 6E, the differences between controls and patients is very small (mainly due to a few outliers) and hardly justifies the suggestion in the last sentence of the results that the increased sAxl would prevent engagement of Gas6-Axl inhibitory pathway in RA DC.

This is a valid point. In order to explore whether the small difference between patients and controls was consistent across different patients' groups, we measured sAXL in an additional cohort of patients (cohort 3, as described in your point 3 above), which now contained a naïve-to-treatment group. Soluble AXL was increased in RA patients naïve-to-treatment and resistant-to-treatment as compared to healthy donors. The highest concentration of sAXL was detected in a naïve to DMARDs group, which was twice that of healthy donors; medians with interquartile range: 29.1 (15.9- 40.2) vs 16.2 (13-22), respectively.

The serum concentrations of sAXL in OA were similar to that of healthy donors and significantly lower than RA patients naïve to DMARDs ($p=0.029$) suggesting that increased levels of sAXL in the sera might be associated with the autoimmune component of RA. Also, in the light of these new data, we suggest that sAXL could indeed interfere with engagement of GAS6 with membrane AXL at least in treatment naïve RA patients. These new data are now presented on Supplementary Figure 8 and corrected conclusion included in the text, pages 11 (line 27) and 12 (lines 1-18).

5. Are the significance indications in Fig 6 D and E the same for all compared groups? Is this a nonparametric test or a Kruskal-Wallis test?

Data presented in Figures 6D-E are now updated and presented in Supplementary Figure 8 A-B. The data are analysed using the Kruskal-Wallis test followed by Dunn's multiple comparisons test. The significance indications were different for each of the patient group as follows: RA naïve-to-treatment $**p=0.0007$, resistant-to-treatment, $*p=0.0042$; patients in remission did not show statistically significant difference; all compared to healthy donors. This has been added to the Supplementary Figure 8 legend.

6. In Supplemental Fig 1, the sorting strategy of the cells is shown. It is not clear, however, why the CD16⁻CD141⁻ population is defined as CD1c⁺ per default.

We apologise for the confusion. We had already CD1c⁺ antibody in our sorting strategy and verified that CD16⁻CD141⁻ cells were CD1c⁺ positive.

We tested many gating strategies and found that exclusion of CD16⁺ and CD141⁺ was the most optimal strategy for sorting pure CD1c⁺ cells. This was true for both blood and synovial tissues/synovial fluid. We also verified the purity of the sorted cells. A more comprehensive sorting strategy is now shown on Supplementary Figure 1, and described in greater detail in the M&M section on page 18, lines 6-26.

7. Many figures lack indications for statistical significance. If the data is analyzed but not

significant, please indicate so in the figure, especially in figures where it is not evident that it is not (e.g. Fig2C or 3H)

All data that are statistical significant are indicated with an asterisk. Otherwise the data is not statistical significant and we included “ns” in the figures where appropriate. This is added to figure legends. The reviewer helpfully identified that the asterisk marking statistical significance was missing in Figure 2C and is now added.

8. Are the stimulations in figure 2 C-D done on the whole monocyte population or on a subpopulation?

This was done on CD14⁺ monocyte population (isolated with anti-CD14 beads; Miltenyi Biotec; purity >96%.) that includes CD14⁺ and CD14⁺CD16⁺ populations. We clarify this in the M&M section on page 17, line 24-25.

We had chosen this approach because these populations give a rise DCs in the inflamed tissue (Segura et al Immunity, 2013, 38:336; Sallusto et al J Exp Med 179:1109).

9. Why do the western blots measuring Axl expression look so different in figure 5 E and F?

The Western blot presented on Figure 5E shows mouse Axl in DCs while the Western blot presented on Figure 5F shows human AXL in DCs. While the protein extracts were prepared with the same method, the blots were developed with different antibodies: goat anti-mouse Axl and rabbit anti-human AXL (M&M, page 22, line 1-10); and these could have contributed to the difference in the appearance of the blots. We clarified which blot shows mouse and which shown human AXL in the Legend for Figure 5 on page 34.

10. Is Gas6-production different between miR-34a^{-/-} and wt CD1c⁺DCs? cells? Is there more Gas6 on the surface of miR-34a^{-/-} CD1c⁺ DCs compared to wt DCs?

This is also a good point. We measured GAS6 concentrations in the culture supernatant (SN) of WT and miR-34a^{-/-} DCs, stimulated with LPS or left unstimulated. We found lower levels of GAS6 in SN of miR-34a^{-/-} stimulated with LPS as compared to WT cells. This could suggest a lower production of GAS6 by miR-34a^{-/-} DCs or an increased binding of GAS6 to membrane Axl on miR-34a^{-/-} DCs, which we found to be upregulated. We speculate that the latter takes place because separate studies conducted by one of the co-authors have shown that in mouse GAS6 is bound to membrane Axl, and that the presence of GAS6 in tissues was dependent on the presence of surface Axl (Zagorska et al. Nature Immunology

2014;15:920-8.). These new data are presented on Supplementary Figure 7A and included in the text on page 11, line 1-5.

11. To confirm that the miR-34a regulated effects on cytokine production and antigen presentation are opposed in Axl KO mice, the authors should perform similar

experiments as in Fig 4 on CD1c+ DCs from Axl KO and wt mice, especially regarding MHC surface expression, DC-T-cell interactions upon OVA challenge and cytokine production.

Axl deficient DCs have been extensively characterised by Prof Lemke; one of the co-authors. He and collaborators have a series of papers showing that DCs from triple-TAM receptor-deficient mice (Axl^{-/-}MERK^{-/-}Tyro3^{-/-}) have high expression of MHC class II and co-stimulatory molecules, produce more cytokines and are less efficient in limiting T-cell activation; and that Axl receptor deficient DCs show an intermediate phenotype. These studies are now cited in our manuscript (page 10, line 7). We had to prioritise experiments to fit the time-limits for re-submission of our revised manuscript so we didn't fully extend this work. Instead we focused on extending the investigation of miR-34a/Axl in additional RA and PsA blood and synovial tissue patients' samples.

Nevertheless, to verify that Axl^{-/-}DCs show an activated phenotype, we performed some additional experiments on Axl^{-/-} DCs that confirmed that in contrast to miR-34a^{-/-} DCs, Axl^{-/-} DCs have a higher expression of MHC class II and produce more pro-inflammatory cytokines, e.g. TNF and IL-6, compared to WT DCs following TLR stimulation. This is consistent with our experiments where neutralisation of AXL by siRNA in miR-34a inhibitor transfected human DCs de-represses cytokine production (Figure 6 A-B). These data are included in the Supplementary Figure 6 and discussed in the text on page 10, lines 7-10.

Minor points:

- 1. Page 3, line 7: woman>women*
- 2. Page 12, line 16: Sentence should start "In RA patients..."*

These have been corrected.

Reviewer #3:

The work is original, and propose identification of a new pathway (miR34a/Axl/Gas6) to understand DC activation and autoimmunity. First, the author identifies miR34a as increased in blood CD1c+ DCs from RA patients than healthy control subjects, and this was further increased in SF DCs. Moreover, miR34 is induced by GMCSF. These points raise some questions:

1. Does inflammatory status or use of anti TNF change expression? Is this specific to RA or find similar results in other chronic inflammatory diseases?

To address this, we investigated miR-34a expression in FACS-sorted CD1c⁺ dendritic cells from a prospective cohort: RA blood (PB) (n=19), RA synovial tissue (n=9), psoriatic arthritis (PsA) PB (n=16), PsA synovial tissue/synovial fluid (n=5); and PB from age-matched control healthy donors (HD) (n=10). The demography of this new CD1c⁺ dendritic cell patient cohort (Cohort 2) is presented in Supplementary Table 2.

The RA patients were either DMARDs treatment naïve (n=7) or resistant-to-treatment (n=12: to methotrexate n=9, and a combination of MTX and TNF inhibitors n=3). This second cohort confirmed the findings in the initial cohort of an increased expression of miR-34a in PB CD1c⁺ dendritic cells from RA patients compared to age-matched healthy control subjects, and revealed a further increase in miR-34a expression in synovial tissue CD1c⁺ dendritic cells (new Figure 1C). The highest fold-change compared to HD was in treatment resistant RA, and in these subjects the miR-34a expression positively correlated with disease duration (r= 0.543, p=0.016).

To clarify if the miR34a/Axl pathway is deregulated in other chronic inflammatory diseases, we tested miR34a expression in CD1c⁺ cells isolated from blood and synovial tissue/fluid of PsA patients. PB CD1c⁺ dendritic cells from PsA patients did not show significant difference in miR-34a expression levels compared to HD. We observed an increase in miR-34a in PsA synovial tissue/fluid CD1c⁺ dendritic cells compared with healthy circulating CD1c⁺ (new supplementary Figure 2); however the upregulation was significantly lower than in RA synovial tissue CD1c⁺ (p=0.02).

In summary, these data confirmed the potential for a strong contribution of miR-34a to dendritic cell function within RA synovial tissue, particularly in patients resistant to treatment. Our data also suggest a potential involvement of miR-34a in driving an immune response in PsA joints however potentially to a lesser degree than in RA. These points are discussed in the text on page 7-8, lines 19-27 and 1-8.

2. In a second step, the authors provide animal experiments and convincing data in miR-34a^{-/-} mice had a reduced incidence and severity of arthritis. However, it is not clear if this effect is dependent only on mir34 expression in DC or if miR34a has other target in immune cells or synovial cells. They show a decrease in Th17 number, is it direct effect on T cells or indirect on DC? but what is the impact on T cells functions? B cells in miR34^{-/-} mice?

Regarding the Th17 cells, we showed that miR-34a supports Th17 development indirectly; through the regulation of DC phenotype. The adoptive transfer of OVA specific naïve WT T cells into WT and miR-34a^{-/-} recipient mice showed a strong reduction in WT Th17 development in miR-34a^{-/-} recipient mice as compared to WT recipient controls upon antigen challenge (Figure 3 G-H; page 8, line 8-15). This suggests that deficiency in miR-34a in DCs (because these are only cells able to prime naïve T cells) inhibits Th17 cell development.

Regarding the lymphocytes in miR-34a^{-/-} mice, we observed no difference in the collagen-specific antibody production between WT and miR-34a^{-/-} mice subjected to collagen-induced arthritis, suggesting that at least in this model, B cell function was not affected by miR-34a deficiency. The new data are now presented in Supplementary Figure 3 and discussed in the text (page 8, lines 5-6).

Apart from an impact of miR-34a^{-/-} deficiency on IL-17 production we found no difference in Th1 and Th2 derived cytokines suggesting no contribution of miR-34a to their function at least in mouse. Another group led by Ventura also found no difference in B and T cell numbers in miR-34a^{-/-} mice compared to WT littermates (Plos Genetics, 2012, 8, e1002797) at least in a steady-state.

3. pathway: they report that microRNA-34a has 416 putative targets, among others bcl2, caspase.....Did the author check apoptosis pathway of DC before focusing on Axl? pro-inflammatory cytokines (TNFa, IL6) was reduced by ex vivo gene-silencing of miR-34a which was restored by down-regulation of AXL suggesting that the control of DC activation is via regulation of tyrosine kinase receptor AXL.

Among the 416 conserved targets of miR-34a, only 43 are potentially expressed in myeloid cells. We selected AXL for our study because it was described as a key regulator of DC activation; and our preliminary functional experiments with AXL siRNA showed the contribution of AXL to the miR-34a-driven DC phenotype. We do not exclude the contribution of other targets, e.g. PDCD4, a regulator of IL-10 (Sheedy et al, Nature Immunology, 2010;11:141-7.) to miR-34a-driven DCs phenotype. Exploring this is likely beyond the scope of the current manuscript and will be investigated in future studies.

To test for apoptotic tendency in miR34a^{-/-} DCs, we evaluated the survival of WT and miR-34a^{-/-} DCs *in vitro*; unstimulated and stimulated with TLR ligands and found no difference in cell survival. We included this survival rate data in the M&M section on page 21, line 19-20.

4. At last, RA patients had higher concentrations of GAS6 and sAXL in the serum independently of their disease activity, but the distribution seems very large in different group of patients, and in methods patient treatment seems heterogeneous. Again, checking in other inflammatory diseases to assess specificity would be relevant. define subgroup of patients according to sAXL level may provide information? Does the level GAS6/sAxl changing overtime?

We recognised the limitation of our clinical dataset. To validate our initial findings, we evaluated serum concentrations of sAXL and GAS6 in an additional prospective (cohort 3), which now contained a naïve-to-treatment RA group and serum from osteoarthritis (OA) patients. The cohort demography and clinical information is presented in Supplementary Table 3, and included

- early RA patients that were naïve to DMARDs treatment (n=20);
- RA patients that were resistant to DMARDs or a combination of DMARDs and TNF inhibitors treatment (n=20);
- RA patients in a stable remission induced by a combination of Methotrexate and TNF inhibitor treatment (n=10);
- healthy donors (n=10)
- osteoarthritis (OA) patients (n=10) as a control for joint disease that does not have an autoimmune component.

Soluble AXL was increased in RA patients naïve-to-treatment and resistant-to-treatment as compared to healthy donors. The highest concentration of sAXL, that also showed tight distribution, was detected in naïve to DMARDs group, which was twice that of healthy donors: medians with interquartile range: 29.1 (15.9- 40.2) vs 16.2 (13-22), respectively. The RA in remission group showed upregulated levels of sAXL compared to healthy donors; however the levels were quite variable between the patients and did not reach statistical significance. The serum concentrations of sAXL in OA were similar to that of healthy donors and significantly lower than RA patients ($p=0.029$). The lack of a clear downregulation of sAXL in RA patients in remission (no inflammation) and low levels of sAXL in OA suggest that levels of serum sAXL in RA could be associated with the underlying autoimmunity per se and to a lesser extent to disease activity measures (inflammation). We included these new data in Supplementary Figure 8 and in the text (pages 10-11, lines 27 and 1-12).

We did not have an opportunity to investigate whether soluble AXL changes with time in sequential patient samples. We plan to investigate this in the future by testing sAXL levels in ACPA positive pre-clinical RA and in sequential blood samples of patients in remission upon withdrawal of treatment and in any of these that subsequently develop flairs of disease. These cohorts are currently being gathered.

This new data is presented on Supplementary Figure 8 and the amended conclusion included in the text (page 11, line 27 and -12, lines 1-18).

REVIEWERS' COMMENTS:

Reviewer #1 (Remarks to the Author):

****No further comments to the authors****

Reviewer #2 (Remarks to the Author):

The authors have adequately addressed all my concerns and revised the manuscript accordingly. I have no further questions or comments.

Reviewer #3 (Remarks to the Author):

the authors have assessed miR34a in blood CD1c+ DCs from RA patients and showed increase compared to healthy control subjects, and this was further increased in SF DCs. Moreover, the target of the miR34a is GAS6/Axl.

They have convincingly demonstrated the data, and answered all questions, performing extra in vivo experiments in AXI ko mice as well as checking in other inflammatory arthritis (as psoriasis arthritis).

Point-by-point replies to reviewer comments

REVIEWERS' COMMENTS:

Reviewer #1 (Remarks to the Author):

No further comments to the authors

Reviewer #2 (Remarks to the Author):

The authors have adequately addressed all my concerns and revised the manuscript accordingly. I have no further questions or comments.

Reviewer #3 (Remarks to the Author):

The authors have assessed miR34a in blood CD1c+ DCs from RA patients and showed increase compared to healthy control subjects, and this was further increased in SF DCs. Moreover, the target of the miR34a is GAS6/Axl.

They have convincingly demonstrated the data, and answered all questions, performing extra in vivo experiments in AXLKO mice as well as checking in other inflammatory arthritis (as psoriasis arthritis).

We appreciate that all three reviewers were satisfied with the improved manuscript.